# Basal Cell Carcinoma: From Pathophysiology to Novel Therapeutic Approaches

**DOI:** 10.3390/biomedicines8110449

**Published:** 2020-10-23

**Authors:** Luca Fania, Dario Didona, Roberto Morese, Irene Campana, Valeria Coco, Francesca Romana Di Pietro, Francesca Ricci, Sabatino Pallotta, Eleonora Candi, Damiano Abeni, Elena Dellambra

**Affiliations:** 1Istituto Dermopatico dell’Immacolata-IRCCS, via dei Monti di Creta 104, 00167 Rome, Italy; r.morese@idi.it (R.M.); irenecamp89@gmail.com (I.C.); cocovaleria@hotmail.it (V.C.); f.dipietro@idi.it (F.R.D.P.); francesca.ricci@idi.it (F.R.); s.pallotta@idi.it (S.P.); candi@uniroma2.it (E.C.); d.abeni@idi.it (D.A.); e.dellambra@idi.it (E.D.); 2Department of Dermatology and Allergology, Philipps University, 35043 Marburg, Germany; dario.didona@gmail.com; 3Department of Experimental Medicine, University of Rome Tor Vergata, Via Montpellier, 1, 00133 Rome, Italy

**Keywords:** basal cell carcinoma, nonmelanoma skin cancer, keratinocyte carcinoma, hedgehog, PTCH1, dermoscopy, therapy

## Abstract

Basal cell carcinoma (BCC) is the most common human cancer worldwide, and is a subtype of nonmelanoma skin cancer, characterized by a constantly increasing incidence due to an aging population and widespread sun exposure. Although the mortality from BCC is negligible, this tumor can be associated with significant morbidity and cost. This review presents a literature overview of BCC from pathophysiology to novel therapeutic approaches. Several histopathological BCC subtypes with different prognostic values have been described. Dermoscopy and, more recently, reflectance confocal microscopy have largely improved BCC diagnosis. Although surgery is the first-line treatment for localized BCC, other nonsurgical local treatment options are available. BCC pathogenesis depends on the interaction between environmental and genetic characteristics of the patient. Specifically, an aberrant activation of Hedgehog signaling pathway is implicated in its pathogenesis. Notably, Hedgehog signaling inhibitors, such as vismodegib and sonidegib, are successfully used as targeted treatment for advanced or metastatic BCC. Furthermore, the implementation of prevention measures has demonstrated to be useful in the patient management.

## 1. Introduction

The incidence of nonmelanoma skin cancers (NMSC), which include basal cell carcinoma (BCC), cutaneous squamous cell carcinoma (cSCC), and actinic keratosis (AK), is increasing worldwide. BCC is the most frequent skin cancer, with constantly increasing incidence due to an aging population and widespread sun exposure. BCC accounts for 50% of all cancers in the United States [1]. Although the mortality from this cancer is negligible, BCC can be associated with significant morbidity, especially if the tumor is untreated for a long period of time [2]. Clinically, it presents with several morphologies and the clinical differential diagnosis should take into consideration a broad variety of diseases ranging from benign lesions to melanoma. Dermoscopy, and, more recently, reflectance confocal microscopy (RCM) have largely improved BCC diagnosis. Several effective therapeutic approaches are available to treat BCC, and an appropriate selection requires knowledge of complications, cosmetic outcomes, and recurrence rates. Additionally, patient’s preferences, of course, must be explored and included in the therapeutic decision. Surgery remains the first-line treatment for this disease. The pathogenesis of BCC depends on the interplay between environmental factors and genetic features of the patient. Aberrant activation of Hedgehog signaling is a driver of BCC development, and its blockade represents a potential therapeutic target. Therefore, Hedgehog pathway inhibitor therapy is emerging as a useful targeted treatment for patients with local advanced or metastatic BCC. Moreover, the implementation of prevention measures may play a crucial role in the improvement of BCC management.

This review provides an overview of BCC from pathophysiology to novel therapeutic approaches.

## 2. BCC Risk Factors

Risk factors for BCC include age, exposure to UV light (including professional and leisure-time exposures), skin phototype, gender, pharmacological therapy, radiation therapy, family history of skin tumors, long-term exposure to arsenic, immunosuppression, and some genetic syndromes.

BCCs are prevalent in the elderly population, given their association with cumulative sun exposure and other exogenous damage. White individuals of old (65–79 years) to very old age (>80 years) display the highest increase in BCC incidence rates [3].

Age-related deterioration of biological functions results in decline of DNA repair capacity, genomic instability, decline of immune system function, and chronic inflammation. Thus, aged skin is characterized by the accumulation of both DNA damage and senescent cells, as well as by the presence of chronic inflammatory status that leads to modifications of dermal matrix integrity. Therefore, some mutant genotypes selectively favor the growth of epidermal subclones in a microenvironment suitable for tumor development created by chronic inflammation [4,5].

Exposure to UV radiation (UVA and UVB wavebands) promotes skin cancer development by direct cell damage, including DNA mutations (e.g., pyrimidine substitutions), induction of oxidative stress, and generation of an “energy crisis” that impairs effective DNA repair, activation of local inflammatory processes, and suppression of cutaneous antitumor immunity [6]. Some studies indicate that intermittent and intense sun exposure (e.g., number of sunburns) increases the risk of BCC development, whereas cumulative and long-term UV exposure does not. In contrast, cSCC risk is mainly associated with cumulative sun exposure during lifetime [7,8]. A population-based, case–control study carried out in Canada reported that childhood and adolescence are the most critical life periods for BCC risk in adulthood [9]. This relation is emphasized among sun-sensitive subjects with tendency to burn. The study also shows a positive association between BCC and Northern European ethnic origin, light skin color, severe sunburns, and freckling in childhood [9]. Moreover, a link between BCC incidence and distance from the equator has been reported. In fact, UV radiation exposure at lower latitudes (e.g., Hawaii) is stronger compared to higher latitudes (e.g., Midwest of the USA) [10,11]. Furthermore, the use of tanning beds, especially early in life, resulted in a 40% increase in risk, compared to a control population. The risk increases in a dose-dependent manner with years of use of indoor tanning devices [12,13].

BCC incidence was generally higher in men than in women probably due to increased recreational and occupational exposure to the sun. However, these differences are becoming less significant with changes in lifestyle, such as tanning bed use or smoking. Gender substantially modifies the age-specific BCC risk. BCC affects preferentially elderly males (>60 years old) and younger female (<40 years old). Notably, women can present BCCs earlier (mainly on the face and neck), because they are more concerned about their health and aesthetics, and go to the dermatologist earlier when they have lesions in these areas [14,15,16].

The BCC risk associated with high doses of psoralen and ultraviolet A (PUVA) therapy, an effective treatment modality for psoriasis, is modest in contrast with the markedly increased risk of SCC at the same high doses of PUVA treatment [17].

Many drugs, such as tetracyclines, thiazide diuretics, nonsteroidal anti-inflammatory drugs (NSAIDs) NSAIDs, and retinoids are potentially photosensitizing, therefore they may induce phototoxic and/or photoallergic cutaneous reactions acting as co-carcinogens with UV radiation, and thus increasing the risk of skin cancer [18]. Experimental and epidemiological findings suggest a link between drug-induced photosensitivity and skin cancer, probably through the induction of DNA damage in predisposed individuals. BCC risk, in particular for the early-onset cancers, is mainly increased by antimicrobials. In fact, tetracyclines are often used for the treatment of acne in adolescence, a life period in which UV exposure has been associated with increased risk for adult BCC [9].

Antihypertensive medications might influence skin homeostasis through different mechanisms. Indeed, some drugs affect epidermal differentiation by interacting with calcium or sodium channels of the skin. Mediators in the renin-angiotensin system (RAS) are also involved in the modulation of cellular proliferation and angiogenesis. Of note, the existence of RAS has been recognized in many organs and tissues, including the skin [19]. Angiotensin receptor blockers (ARBs) users have an increased BCC risk in comparison to ACE inhibitors users because ARBs selectively stimulate type 2 receptors instead of type 1 ones, enhancing angiogenesis and cancer progression. A Dutch study found an increased risk of BCC among long-term users of loop diuretics, without any association with thiazides and potassium-sparing agents [20]. Nevertheless, the poor available data regarding the relationship between use of different types of antihypertensives and skin cancer risk do not allow to draw definite conclusions yet. Additionally, the consumption of arsenic-contaminated water and arsenic-containing medications, are associated with the development of BCCs [21,22].

Ionizing radiation exposure in environmental, occupational, and therapeutic settings increases risk of BCCs but not of other skin cancers. Overall, this risk is higher in Caucasians than other races. Several studies reported higher incidence of radiation-induced scalp BCCs in children irradiated for tinea capitis [23,24,25]. An inverse relationship between BCC risk and age at radiation therapy exposure was observed [23,26]. Notably, the infiltrative subtype of BCC, that is considered to be more aggressive, was significantly more frequent in irradiated patients [27].

Furthermore, previous history of BCC represents a risk factor for additional skin cancers, including NMSC and melanoma. Many of these patients (from 30% to 50%) will develop another BCC within 5 years [28,29,30,31]. These patients display a 10-fold risk increase compared to the general population [29]. A prospective cohort study on 1426 patients indicates that 40.7% (95% CI, 36.5–45.2%) of them develop a new NMSC within 5 years after the first lesion and 82% (95% CI, 80.2–83.7%) of them develop a new NMSC within 5 years after more than one lesion [32].

Another prospective population-based cohort study confirmed that development of a second BCC is most likely during the short-term follow-up period after diagnosis of the first lesion [33]. A recent Italian study found that NMSC patients had a relative risk (RR) for melanoma of 6.2 compared to controls. Melanoma risk was particularly high in patients who had NMSC before the age of 40 (RR of melanoma = 25.1 compared to controls) [34].

Immunosuppression in organ transplant recipients increases the risk of NMSC and the increase depends on the duration of the immunosuppressive therapy. Notably, BCCs incidence increases 10-fold in transplant recipients. HIV seropositivity doubles the risk of BCCs [21].

Several genetic syndromes are associated with BCCs development. The most common is the basal cell nevus syndrome (BCNS) or nevoid basal cell carcinoma syndrome or Gorlin syndrome, characterized by multiple BCCs development in childhood, most commonly in the face, followed by chest, back, and scalp. Other typical manifestations include palmar pits, jaw cysts, ectopic calcifications in falx cerebri, skeletal abnormalities, characteristic facial appearance, ovarian fibromas, medulloblastomas, and meningiomas. Other syndromes include multiple hereditary infundibolocystic BCC, Rombo syndrome, Bazex–Dupré–Christol syndrome, epidermolysis bullosa simplex, Dowling-Meara, and albinism [35].

## 3. Clinical Features and Different Subtypes of BCCs

Several clinical subtypes of BCC have been described in the literature, but the main clinical subtypes are nodular, superficial, and morpheaform BCC [21,36]. A variable amount of melanin can be present in case of pigmented BCC (Figure 1) [21,36]. Usually, BCC occurs on the face and hairy skin, including the upper and lower extremities [21,36]. More rarely, BCC involves the genital mucosa [21,36].

Nodular BCC (NBCC) is the most common clinical subtype, accounting for 50–79% of all BCCs [37,38]. NBCC appears as a papule or nodule with a characteristic pearly, shiny edge, and small arborizing telangiectasias (Figure 2). The lesion may ulcerate, but a sharp border usually is maintained, which can be a clue to the diagnosis [37]. NBCC involves most frequently the face, especially the cheeks, nasolabial folds, forehead, and eyelids. Several differential diagnoses should be considered, including molluscum contagiosum, sebaceous hyperplasia, amelanotic melanoma, trichoepithelioma, and Merkel cell carcinoma. Ulcerated lesions can be misdiagnosed for cSCC or keratoacanthoma.

Superficial BCC is the second most common clinical subtype of BCC, accounting for up to 15% of BCC [37,38]. It is characterized by a sharply circumscribed, scaly, pinkish macule, papule, or thin plaque (Figure 3). When a spontaneous regression occurs, it typically leaves atrophic and hypopigmented areas. Superficial BCC usually involves the trunk and extremities, and among younger patients may be more frequent than the other subtypes [39]. The differential diagnosis includes AK, Bowen’s disease, lichenoid keratosis, guttate psoriasis, and nummular eczema. Superficial BCCs are frequently multifocal, what can lead to an incomplete excision [39]. Therefore, an accurate follow-up should be performed [39].

Morpheaform BCC (MBCC) accounts for up to 10% of BCC [38]. Its name is due to its clinical resemblance to an indurated plaque of localized scleroderma. It is usually characterized by ivory-white, shiny, smooth, indurated plaques or depressions with poorly defined edges (Figure 4). MBCC is usually more aggressive than other forms of BCC and it usually spreads subclinically, leading to extensive local destruction. MBCC may be misdiagnosed for a scar, localized scleroderma, dermatofibrosarcoma protuberans, Merkel cell carcinoma, or amelanotic melanoma.

Only 1% of BCC evolves in a giant BCC or in ulcus rodens, an extremely destructive form of BCC that shows deep tissue invasion and a high rate of postsurgical recurrence (Figure 5) [40]. Furthermore, combinations of these subtypes may be found in a single lesion, which is then referred to as a mixed tumor, that account for approximately 40% BCCs [41].

According to prognostic factors, BCC can also be classified into low risk and high risk [42]. These prognostic factors are tumor size (larger size leads to a higher risk of recurrence), definition of clinical margins (poorly defined lesions are at higher risk), histological subtype (morpheaform, and metatypical BCC represent high risk lesions), histological features (perineural and/or perivascular invasion are markers of higher risk), recurrence, and tumor location [42]. Regarding the location, high-risk zones are represented by the nose, periorificial areas of the head and neck; intermediate-risk zones are the forehead, cheek, chin, scalp, and neck; low-risk zones are the trunk and limbs. High-risk BCCs are characterized by one poor prognostic factor at least; while, low-risk BCCs are superficial BCC, Pinkus tumor (a variant of BCC characterized by focal cystic changes), and small nodular BCC on intermediate or low-risk areas [42]. French guidelines defined also an intermediate prognosis group, to separate recurrent superficial BCC from other recurrent BCC, and some NBCC according to size and location (Table 1) [43]. Furthermore, all BCCs managed by ablative procedures (i.e., laser therapy) without histopathological control instead of surgical excision are at high risk of recurrence [44].

## 4. Histopathological Features of BCC

BCC derives from the basal cell layer and outer root sheath of the hair follicles, which contain pluripotent epithelial cells [1,45]. It was postulated that the lack of cytokeratin 15 in BCC suggests a derivation from the bulge region of the hair follicle [46]. Furthermore, the expression of CD10 emphasizes the follicular derivation of these epithelial cells [47,48]. Several histopathological subtypes of BCC, with different prognostic values, have been described in the literature [45,49]. 

NBCC is the most common subtype of BCC [49]. NBCC is characterized by nests of basaloid cells with sharp borders, showing a characteristic peripheral palisading of cells and a typical cleft (Figure 6) [49,50]. The presence of bulky aggregates of mucin can produce a cystic structure [1]. Calcification can also be detected, especially in long-standing lesions [45]. Mitotic activity is usually not visible, but a high mitotic rate is reported as a feature of more aggressive NBCC (Figure 7) [45,49]. Adenoidal BCC is considered a variant of NBCC, characterized by nests of basaloid cells extending into the dermis (Figure 8) [51]. Another variant of NBCC is the basal cell epithelioma, characterized by enlarged, mononuclear, and/or multinucleated cells, also known as monster cells [52]. Furthermore, several other subtypes of NBCC have been reported, including granular cell BCC, and BCC with outer hair follicle sheath differentiation [45,49].

Micro-nodular BCC is clinically represented by a plaque and is characterized by an increased recurrence rate [53]. Pathologically, it shares several features with NBCC, but micro-nodular BCC is smaller, and it is characterized by micro-nodules of basaloid cells, minimal palisading, and myxoid surrounding stroma [53].

Superficial BCC is another common variant of BCC, characterized by nests of basaloid cells that extend from the epidermis and a prominent palisading [49].

The infiltrative variant of BCC is a continuum between NBCC and MBCC, characterized by nodules of atypical basaloid cells with different sizes and surrounding mucinous stroma (Figure 9) [45,49,54]. This subtype may also involve the subcutaneous tissue and the adnexa [45,49,54]. Therefore, this subtype of BCC is more aggressive, and its surgical eradication is not simple [45,49].

The NBCC infiltrative type consists mainly of nests of basaloid cells that usually infiltrate the collagen fibers. In contrast to the MBCC, the stroma is not compact and shows plenty of mucine [55]. Furthermore, NBCC nests of basaloid cells are accompanied by a poorly demarcated peripheral palisading and show widespread invasion of the reticular dermis and even penetration into the subcutaneous fat.

MBCC shows thin and elongated islands of neoplastic cells not arranged in nodules [56]. In MBCC, the peripheral palisading is, in contrast in to NBCC, usually completely absent, and the stroma retraction, characterized by retraction spaces that mimic vascular invasion, is also infrequent MBCC may also involve the subcutaneous tissue and the adnexa [56].

Metatypical BCC is characterized by features of nodular BCC and cSCC of the skin [57]. In this variant, basaloid cells show variable eosinophilic features, prominent mitotic activity, and numerous apoptotic cells [57]. Furthermore, peripheral palisading is not always present [44,57]. Metatypical BCC may show perineural and lymphatic involvement [57].

The term ‘‘basosquamous carcinoma’’ is referred to a BCC with areas of lineage differentiation into squamous cell carcinoma [58,59]. It shows areas of BCC, cSCC, and a transition zone between these entities [58,59]. In contrast to the metatypical BCC, the term basosquamous carcinoma should be used in the case of tumors with contiguous areas of BCC and SCC [59].

On the one hand, the areas of BCC are characterized by small, uniform, hyperchromatic cells, with peripheral palisading, mitoses, and stromal collagen deposition with proliferative fibroblasts [60]. On the other hand, the areas of squamous-like cell carcinoma are characterized by large polygonal squamous cells with eosinophilic cytoplasm, large nuclei with prominent nucleoli, and frequent mitosis [60].

## 5. Dermoscopy

Dermoscopy is a noninvasive in vivo technique that significantly improves the early diagnosis of melanoma and NMSC. It allows an examination of pigmented and nonpigmented skin lesions. The accuracy of dermoscopy for the diagnosis of BCC has been widely reported in the literature [61,62]. The clinical recognition of both types of BCC, pigmented and nonpigmented, is significantly increased by many dermoscopic criteria [63,64,65]. The diagnostic accuracy of dermoscopy for BCC could range between 95% and 99% and this could depend on BCC type compared to other lesions in the control group, such as melanocytic and nonmelanocytic lesions [61,62,66].

Dermoscopy structures of BCC consist of three categories that include vascular, pigment-related, and nonvascular/nonpigment-related (Table 2). Vascular structures are arborizing vessels and short fine telangiectasias while pigmented-related structures include maple leaf-like areas, spoke-wheel areas, multiple blue–grey nests and globules, in-focus dots, and concentric structures (Table 2). Nonvascular/nonpigmented structures include ulcerations, multiple small erosions, shiny white–red structureless areas, and white streaks (Table 2).

Starting from the dermoscopic vascular features, arborizing vessels are the main structures in BCC, with a high diagnostic accuracy and a positive predictive value of 94.1% [62,67,68,69]. Arborizing vessels consist of stem vessels of large diameter, characterized by a bright red color, branching irregularly into finest terminal capillaries (Figure 10a) [64,70,71]. Short fine telangiectasias are vessels with a small diameter and length of <1 mm, with few or no branches (Figure 10b) [70]. With regards to the pigmented-related structures, blue–grey ovoid nests are confluent or nearly confluent, well-circumscribed, pigmented ovoid, or elongated areas (Figure 11a) [61]. Maple leaf-like areas are bulbous extensions connected at a base area, usually brown or grey–blue in color, forming a leaf-like pattern (Figure 11b) [64,65]. Ulceration is a shallow erosion of the epidermis that reaches the dermis and may be covered with coagulated blood or serous crust (Figure 12a). Multiple small erosions are smaller compared to ulcerations, generally seen as small brown–red to brown–yellow crusts (Figure 12b) [65,70]. White–red structureless areas represent diffuse dermal fibrosis or fibrotic tumor stroma and appear as areas of white to red color (Figure 13a) [65,67,70]. White shiny streaks or chrysalis or crystalline structures can be seen only with polarized dermoscopy as orthogonal short and thick crossing lines and represent dermal fibrosis (Figure 13b) [72,73]. Multiple blue–grey dots and globules are numerous, loosely arranged round to oval well-circumscribed structures, smaller than blue–grey ovoid nests (Figure 14a). In-focus dots correspond to loosely arranged well-defined small grey dots, which appear sharply in focus (Figure 14b) [64]. Spoke-wheel areas are radial projections surrounding a central darker point, blue or grey color, rare in dermoscopy but highly specific for BCC (Figure 15a) [61,64]. Concentric structures are irregularly shaped globular-like structures with different colors, including blue, grey, brown, and black, with a darker central area, that could be variations or ‘precursors’ of the spoke wheel areas (Figure 15b) [64].

The overall dermoscopic pattern of BCC depends on several combinations of the above-mentioned criteria. Several factors may determine the dermoscopic aspects of BCC that could depend on the patient (i.e., gender, age, race, pigmentary traits) or on the tumor (i.e., histopathologic subtype, anatomic site, presence of pigmentation) [64].

## 6. Reflectance Confocal Microscopy and Optical Coherence Tomography

Novel noninvasive approaches for the diagnosis of BCC include RCM and optical coherence tomography (OCT). RCM, using near infrared laser, is able to provide high magnification images of a given skin lesion at a cellular resolution that is similar to those of histopathology but in real time without skin biopsy [74,75]. Several articles extensively described the RCM features of BCC [76,77,78,79,80,81,82,83,84,85]. The most important RCM criteria for the diagnosis of BCC come into view at the level of the superficial dermis or dermal-epidermal junction (DEJ). These are the dark silhouettes (hypo-reflective areas at the level of DEJ or superficial dermis outlined by bright collagen bundles), the bright tumor islands (round to oval, cord-like or lobulated bright structures, often demarcated by surrounding dark cleft), cleft-like dark spaces (black areas, shaped like clefts/slits separating the bright tumor islands to the dermis), dendritic cells (bright delicate, dendritic structures within bright tumor islands or in epidermis), plump-bright cells (oval to stellate cells without nucleus in the dermis), and canalicular vessels (dilated vessels) (Figure 16). RCM in association with dermoscopy can help to diagnose BCC subtypes without skin biopsy [86]. Limitations of RCM include the imaging depth (250 µm), the limited ability to evaluate tumor invasion and deep margins, the initially steep learning curves and the cost of the instrument that represent a barrier to commercial implementation.

OCT is a noninvasive real-time diagnostic assessment of skin that utilizes infrared light projected onto the skin to produce an image based on the sum of light refractions of various skin structures with different optical properties [87]. It has been demonstrated in a cohort study that OCT had a sensitivity and specificity for superficial BCC diagnosis of 87% and 80%, respectively [88]. Furthermore, OCT in association with dermoscopy showed the highest accuracy (87.4%) [87,89]. Limitations of OCT are the lack of Current Procedural Terminology codes and the minimal use for pigmented lesions because, in pigmented tumors, imaging techniques based on the penetration of light is more difficult. Other noninvasive techniques for the diagnosis of BCC, not generally used in daily practice, include Raman spectroscopy (a spectroscopic technique typically used to determine vibrational modes of molecules), high-resolution ultrasonography, and terahertz pulse imaging (situated in the frequency regime between optical and electronic techniques) [90,91,92,93].

## 7. Pathogenesis of BCC

The development of a BCC results from the interaction between several genes and environmental factors. Notably, most of genes involved in BCC pathogenesis display a mutational signature consistent with UV-induced DNA damage [94,95]. Since BCCs display a great variability in morphology, aggressiveness, and response to treatment, unraveling the molecular genetics of BCC pathogenesis may improve the development of novel targeted therapies to enhance treatment efficacy and overcome tumor resistance.

### 7.1. Canonical Hedgehog (HH) Pathway Genes

Aberrant activation of Hedgehog (HH) signaling is the hallmark of BCC pathogenesis [96]. The HH pathway is a highly conserved signaling pathway that plays a critical role in embryogenesis, cell differentiation, and cell proliferation [97,98]. During embryogenesis, HH signaling orchestrates the morphogenesis of the epidermis and its appendages by signal cross-talk between epithelial and dermal cells. HH is also responsible for maintaining bulge stem cells and controlling the growth of postnatal hair follicles and epidermal renewal [97,98].

“Canonical” activation of HH pathway occurs through binding of the family of extracellular HH ligands (i.e., sonic hedgehog (SHH), Indian hedgehog (IHH) and desert hedgehog (DHH) in mammals) to the 12-pass transmembrane receptor Patched 1 (PTCH1). HH-PTCH1 binding represses the functions of tumor suppressor of PTCH1 allowing the release of the seven-pass transmembrane G-protein coupled receptor Smoothened (SMO). Active SMO migrates to the primary cilium, a highly specialized microtubule-based organelle that acts as a sensor for extracellular signals. Thus, SMO drives a signaling cascade that leads to release and activation of the GLI family of transcription factors (GLI1, GLI2, and GLI3) sequestered in the cytoplasm by several proteins, including the Suppressor of Fused (SUFU). In fact, in the absence of HH ligand, PTCH1 blocks the migration of SMO in the primary cilium and GLI proteins are in their partially cleaved inactive form induced by phosphorylation and ubiquitylation. The translocation of SMO in primary cilium prevents the proteolytic process and the full-length GLI active form moves into nucleus and promotes the transcription of target genes [96,99,100,101]. GLI1 acts as an activator of transcription, whereas GLI2 and GLI3 display both positive and negative transcriptional functions. The HH target genes include GLI1, which further amplifies the initial HH signaling at transcriptional level [102], and PTCH1 and HH interacting protein (HHIP1), which both provide a negative feedback dampening the pathway (Figure 17A) [96,97,98,99,100,101]. The outcome of the HH signaling depends on several of cell-specific targets mediating different cellular responses: proliferation and differentiation (Cyclin D1 and D2, E2F1, N-Myc, FOXM1, PDGFRα, IGFBP3 and IGFBP6, Hes1, Neogenin), cell survival (BCL-2), self-renewal (Bmi1, Nanog, Sox2), angiogenesis (Vegf, Cyr61), cardiomyogenesis (MEF2C), epithelial–mesenchymal transition (Snail1, Sip1, Elk1, and Msx2), and invasiveness (Osteopontin) [96,97,98,99,100,101].

Most components of the HH pathway can function as tumor suppressor or proto-oncogenes, as mutations in their genes promote an oncogenic signaling and are associated with a wide variety of tumors, such as those of the brain, breast, gastrointestinal tract, lung, pancreas, prostate, ovary, and skin, including BCC [96,100] (Figure 17B).

Loss of one functional copy of tumor suppressor *PTCH1* in patients with BCNS (or Gorlin syndrome) predisposes them to BCC development. Most sporadic BCC (85%–90%) have loss-of-function mutations in *PTCH1*, which prevent the repression of HH signaling. About half of *PTCH1* mutations contains the “UV-signature” with C > T and tandem CC > TT transitions; however, other factors, such as oxidative stress, have been implicated in the mutagenesis of this gene [100]. The gain-of-function mutations in the proto-oncogene *SMO*, which become resistant to inhibition by *PTCH1*, can contribute to 10–20% of sporadic BCC development [103,104,105,106]. Simultaneous *PTCH1* and *SMO* mutations have also been observed [107]. Mutations in *GLI1* [108] and *GLI2* [109,110], and those in *SUFU* [111], which disrupt its binding to GLI, have been found in sporadic BCC and can lead to constitutive pathway activation. *PTCH2* gene, which shows a 57% of similarity with its homolog *PTCH1*, has been found to carry mutations in a small number of sporadic BCC [112]. *SHH* mutations are very rare in sporadic BCCs [107]. However, significant levels of *SHH* mutations have been found in patients suffering from Xeroderma Pigmentosum, a rare genetic disease caused by mutations in genes of DNA damage repair systems and characterized by an extreme sensitivity to UV rays and high incidence of BCC [113].

Several studies have demonstrated a tumor-driving role of activated HH signaling in BCC development. Mice overexpressing SHH in the context of normal PTCH develop multiple BCCs and typical features of patients suffering from BCNS [114]. Moreover, the tumor suppressor role of PTCH1 has been demonstrated in *PTCH1*+/− mice that develop tumors and several features observed in patients with BCNS [115,116]. The oncogenic role of SMO has been demonstrated using transgenic mice overexpressing mutant *SMO* that developed skin abnormalities similar to BCCs [103,117]. Overexpression of GLI proteins in mouse models induces BCC development [108,110]. Notably, continued SHH signaling is required for BCC carcinogenesis because mice conditionally expressing GLI-2 show BCC regression when GLI-2 expression is inactivated [118]. Based on these preclinical studies, novel BCC therapies aimed at inhibiting HH-GLI signaling have been focused on the development of SMO antagonists, such as vismodegib and sonidegib, and GLI antagonists [119].

Although HH pathway dysregulation alone drives BCC formation, a complex genetic network of cancer-related genes and different regulatory pathways contribute to BCC development, supporting a heterogeneous genetic origin, and can underlie the pathogenesis of both slow-growing as well as locally invasive BCCs into a rare metastatic disease.

### 7.2. Noncanonical HH Pathway Genes

The HH pathway interacts with various other oncogenic signaling networks, such as EGFR, IGF, TGFβ, aPKC, PI3K, and NF-κB, which synergistically may contribute to BCC development. Indeed, the transcription or post-translational modification of GLI can occur via alternative pathways that are generically defined “Noncanonical” HH signaling, given that the canonical HH-PTCH1 binding and activation of SMO are bypassed [96,99,119]. GLI activity has been shown to be regulated positively by RAS, TGFβ, PI3K/AKT, NF-kB, and aPKCι/λ, and negatively by p53 and PKA (Figure 17C). Specifically, EGFR signaling through the RAS/RAF/MEK/ERK cascade modulates expression of GLI downstream targets by activating JUN/AP-1 complex that acts in cooperation with GLI itself. Moreover, ERK pathway prevents GLI2 proteasome-mediated degradation [120]. TGFβ signaling upregulates GLI2 transcription whereas activation of aPKC phosphorylates and activates GLI1 [121,122]. Stimulation of PI3K/AKT by IGF-1 induces GLI nuclear localization and transcriptional activity [123]. Notably, HH signaling regulates metastasis through the activation of PI3K/AKT, which promotes epithelial–mesenchymal transition and matrix metalloproteinase 9 secretion. NF-κB is a transcription factor triggered by chemicals or UVB, and associated with cutaneous inflammation and carcinogenesis [124]. NF-κB promotes GLI activation by binding to its promoter [125]. p53 represses GLI1 activity, nuclear localization, and transcription levels following DNA damage [126,127]. PKA regulates GLI localization and inactivation through direct phosphorylation. Moreover, activated PI3K/AKT stabilizes GLI2 through inhibition of PKA-mediated phosphorylation [123].

Noncanonical GLI regulation by oncogenic signaling may explain in part the failure of some clinical trials with SMO antagonists. For instance, vismodegib-resistant BCCs show high level of aPKCι/λ, and its pharmacological inhibition by PSI treatment suppresses HH signaling and growth of resistant BCC cell lines [128]. Thus, therapeutic strategies able to inhibit noncanonical HH signaling in combination with SMO or GLI antagonists might reduce resistance mechanisms. Interestingly, the topical treatment imiquimod enhances PKA activity with consequent GLI phosphorylation and cleavage into its repressor form [129].

In sporadic BCCs, inactivating mutations in the *TP53* gene have been found in BCCs (50%) [21]. Most TP53 mutations display an “UV-signature” [100]. Notably, a lower level of *TP53* mutations has been found in BCCs from sunscreen users compared to that of non-sunscreen users [130]. A smaller number of sporadic BCCs displayed mutations in the *CDKN2A* locus, in *RAS* gene family (H-RAS, K-RAS, and N-RAS) and in genes encoding proteins of the PI3K/AKT pathway [21].

### 7.3. Other Genetic Changes

Extensive genomic studies identified novel mutations both downstream of GLI and independently of the HH pathways, which may play a key role in the development and/or progression of BCC [96,101,119,131].

#### 7.3.1. Hippo-YAP and WNT Signaling Genes

Inactivating mutations in *LATS1/2* and *PTPN14* genes, two key components of the Hippo pathway, have been found (Figure 17D) [131]. The Hippo pathway plays a key role in organ size control, and its deregulation contributes to tumorigenesis [132]. The major downstream effector of this pathway is the Yes-associated protein (YAP), a co-transcriptional activator that maintains basal epidermal progenitors, regulates hair follicle growth, and promotes skin proliferation. Upon Hippo signaling activation, YAP becomes phosphorylated, translocates into the cytoplasm, and is sequestered by 14-3-3σ. Thus, it can no longer transcribe its target genes. LATS1/2 kinases prevent the translocation of YAP1 [133]. Thus, mutations in Hippo pathway components induce nuclear localization of YAP and aberrant transcriptional activation promoting BCC development. In addition, interactions between the p53 family of transcription factors and Hippo pathways in subset of physiological contexts, including transformed cells, is responsible for maintaining homeostasis between stemness and differentiation [134,135,136]. Therefore, alterations in this balance may be used by cancer cells to maintain high proliferation features and to increase their competitive advantage also in BCC [137].

Conditional deletion of *YAP* and its paralog *TAZ*, in mouse models of BCC and SCC, prevents tumor formation [138]. *LATS1* gene has been investigated in a study which reported inactivating mutations in 47 of 293 (16%) BCCs, with 24% of truncating mutations, consistent with a tumor suppressor role [131]. Moreover, missense mutations in the *LATS2* gene (12%) and in the *PTPN14* gene (23%) have been found in BCCs. Most of genetic changes (61%) are truncating mutations [131]. In keeping with these findings, the Hippo–YAP pathway is significantly upregulated in BCC [131].

WNT signaling pathway plays a critical role in embryonic development and hair follicle growth and can cross-talk with HH pathway. Notably, WNT signaling initiates hair bud formation, whereas HH signaling promotes the proliferation of follicle epithelium to form a mature follicle. Canonical WNT signaling is activated through binding of WNT proteins (e.g., Wnt-3a) to dimeric receptors composed of the seven transmembrane frizzled (FZD) proteins and the LRP5/6. Upon ligation to receptors, the cytoplasmic protein disheveled (Dvl) is recruited, phosphorylated, and activated. Activation of Dvl induces the dissociation of GSK-3β from Axin leading to GSK-3β inhibition and, in turn, inhibition of phosphorylation and degradation of β-catenin. Thus, β-catenin translocates into the nucleus leading to changes in different target gene expressions. Noncanonical WNTs, including Wnt5a, bind FZD receptors in conjunction with alternate co-receptors, including ROR1/2 or Ryk, inducing β-catenin-independent changes such as PKC activation and cytoskeletal rearrangements [139].

WNT pathway activation has been identified in BCCs. Indeed, BCCs exhibit overexpression of canonical and noncanonical WNTs [140,141] and β-catenin harboring stabilizing mutations [142]. Nearly 30% of BCC display nuclear localization of β-catenin (Figure 17D) [143,144,145]. The canonical WNT signaling pathway is required for HH pathway-driven development of BCCs in a BCC mouse model. Aberrant HH signaling activation induces the Gli-mediated transcriptional activation of *WNT* genes [146]. *LGR4* and *LGR5*, which are target genes of the Wnt signaling pathway and are considered epithelial stem cell markers, are upreguated in BCCs. Moreover, GLI1 and GLI2 expression has strong correlation with LGR4 and LGR5 in BCC from patients and GLI1-expressing stem cells co-express LGR5. Interestingly, a quiescent tumor cell population expressing LGR5 persists following vismodegib treatment in different mouse models and human patients, promoting BCC relapse following treatment discontinuation. These persistent tumor cells present residual WNT signaling activity and could be eliminated by dual WNT and HH inhibition leading to BCC eradication [147].

#### 7.3.2. N-MYCN/FBXW7 Genes

*N-MYC* missense mutations have been identified in 30% of BCCs [131]. MYC family members of transcription activators are involved in regulating embryonic development and cellular mechanisms, such as cell growth, proliferation, differentiation, and apoptosis. Moreover, MYC is a potential downstream effector of the HH pathway [148]. Most of the identified mutations compromise the interaction of N-MYC with the tumor suppressor FBXW7, a component of the SCFFbw7 ubiquitin ligase that promotes proteasome-dependent MYC degradation. Indeed, these mutations enhance N-MYC stability [131]. Moreover, deleterious mutations and LOH events in the *FBXW7* gene have been found in 5% and 8% of BCCs samples, respectively [131].

#### 7.3.3. NOTCH Signaling Genes

NOTCH is a p53 target and a key regulator of epidermal differentiation which becomes activated when normal keratinocytes lose contact with the basal membrane [149]. Human mutations are mostly related to loss of function, indicating a tumor-suppressor role of *NOTCH* in BCC pathogenesis (Figure 17D) [149]. Deep sequencing studies have recently found that mutations in *NOTCH1/2* are among the most common genetic alterations in BCC, although they cannot be considered oncogenic drivers [131,150,151]. Indeed, *NOTCH1*-deficient mice can develop SCCs, and occasionally BCCs [152]. NOTCH inhibition promotes tumor persistence in *PTCH1* conditional mice, whereas NOTCH activation is sufficient to induce regression of established lesions.

#### 7.3.4. TERT-Promoter

A high frequency of mutations has been identified in *TERT* promoter in BCCs [153,154]. The *TERT* gene encodes the catalytic reverse transcriptase subunit of telomerase which maintains telomere length. BCCs with *TERT* promoter mutations display high transcription levels of catalytic subunit [155]. Increased telomerase activity is a hallmark of cancer. Enhancement of TERT transcription is the principal cause of its cancer-specific activation and studies on several tumors support the concept that noncoding mutations within promoter regions of TERT may act as a driver [154]. Various carcinogenic factors (UV or X-ray radiation) can induce distinct “mutation signatures” in *TERT* promoter in BCC [156]. TERT promoter mutations are not associated with clinicopathological features of BCC [156]. Interestingly, the alterations in *PTCH1*, *TP53*, and *DPH3* promoter occur more frequently in BCCs with *TERT* promoter mutations [157,158]. Cancers harboring *TERT* promoter mutations are often more lethal. Some BCCs metastasize resulting in patient mortality. Although these patients can be treated SMO inhibitors, some of them respond with a low rate. Probably, the *TP53* and *TERT* promoter mutations could impact the treatment outcome [158]. Notably, TERT promoter mutations associate with epithelial-to-mesenchymal transition (EMT) gene expression signature and MAPK signaling in several cancers [155].

#### 7.3.5. DPH3-OXNAD1 Bidirectional Promoter

Noncoding mutations in the bidirectional promoter of both *DPH3* and oxidoreductase NAD-binding domain containing 1 (*OXNAD1*) genes have been reported in BCCs [131]. Typical “UV signature” mutations have been found [159]. *DPH3* gene is required for the synthesis of diphthamide, a modified histidine residue in translation elongation factor 2 that is involved in the maintenance of translation fidelity. Its silencing impairs melanoma metastasis in a mouse model, suggesting a role as tumor suppressor [160].

#### 7.3.6. Other BCC-Associated Genes

A high frequency of mutations significantly associated with BCC tumorigenesis was observed in two cancer-related genes, *PPP6C* and *STK19* [131]. PPP6C encodes a phosphatase which regulates cell cycle progression controlling cyclin D1 and pRb inactivation and participating to LATS1 activation [161]. Mutations in *PPP6C* have been detected in 15% of BCC and most of them impair its phosphatase activity [100]. The *STK19* gene encodes a kinase, probably involved in transcriptional regulation. Mutations have been found in 10% of BCCs [100].

#### 7.3.7. Noncoding (nc) RNAs

MicroRNAs (miRNAs), small regulatory RNAs that are involved at multiple transcriptional, post-transcriptional, and epigenetic levels, modulate several processes, and are commonly misregulated in cancer. Altered miRNAs expression levels are associated with BCC progression, suggesting the role of ncRNA regulation in tumor promotion. The expression levels of miRNA machinery (Drosha, DGCR8, AGO1, AGO2, PACT, and TARBP1) have been shown to be significantly higher in BCC when compared to healthy controls [162]. Distinctive miRNA profiles correlate with nodular and infiltrative tumor BCC subtypes. For instance, miR-183, a miRNA that inhibits metastasis in other cancers, displays a significantly lower expression in infiltrative than in nodular BCCs [163]. Moreover, dysregulated miRNA profiles can be associated with some of the key players in BCC pathogenesis. In nodular BCC, upregulated miR-141, 200a, and 200c are linked to C-MYC and the WNT-β-catenin pathway [163,164]. MiR-203 and miR-451a can function as tumor suppressors. HH and EGFR signaling suppresses miR-203 which acts on c-JUN and, in turn, cell proliferation. Notably, in vivo delivery of miR-203 in a BCC mouse model results in the reduction of tumor growth [165]. Additionally, miRNA-451a is significantly reduced in human and mouse BCC. Inhibiting miRNA-451a in primary cells promotes cell growth through its target, TBX1. Conversely, overexpression of miRNA-451a in tumor cells induces cell-cycle arrest, suppressing cell growth [166]. OncomiR-1 cluster (miR-17-92) displays a regulatory role in SHH pathway in a PTCH1 mouse model and corresponding miRNAs are overexpressed in human BCCs [167]. Some long ncRNA (lncRNAs), such as ANRIL, are differentially expressed in BCCs [168].

### 7.4. Cellular Origin of BCC

BCC is so named because it is composed of cells histologically resembling basal keratinocytes of the hair follicle and interfollicular epidermis. However, its cellular origin has been debated for a long time. The advances in understanding molecular genetics of BCC allowed to address this issue. In a mouse model, constitutively active SMO mutant induced BCCs when conditionally expressed in basal keratinocytes of the interfollicular epidermis, but not when expressed in hair follicle stem cells [169]. Cell fate mapping, using cell fate tracking X-ray-induced BCCs in *PTCH* (+/−) mice, demonstrates the origin of BCC from keratin 15-expressing stem cells of the follicular bulge. However, conditional p53 loss produces BCCs from the interfollicular epidermis and it enhances BCC carcinogenesis from the bulge, at least in part by enhancing SMO expression [170,171]. These findings indicate that BCCs may arise from basal keratinocytes of the interfollicular epidermis or of the hair follicle, depending on the context. Induction of GLI2 activator (*GLI2ΔN*) in mice promotes the development of BCC-like tumors whose clinico-pathologic type depends on the cell of origin, tissue context (quiescent versus growing hair follicles), and level of oncogenic signals. Nodular BCC-like tumors arise from hair follicle stem cells, whereas superficial BCC-like tumors arise from interfollicular epidermis, following high level signaling [172]. Only interfollicular epidermal stem cells, and not their committed progeny, are able to transform into BCC upon HH signaling due to their enhanced self-renewing ability. Oncogene-targeted stem cells are characterized by enhanced self-renewing ability and resistance to p53-mediated apoptosis which result in rapid clonal expansion and progression into invasive tumors [173]. Finally, tumor microenvironment may exhibit cancer cell-specific regulatory mechanisms acting on keratinocyte HH pathway and BCC development [174].

## 8. Treatment of Localized BCC

The primary goal for the treatment of localized BCC is the complete excision of the skin tumor and the preservation of the cosmetic and functional aspects. Therefore, surgery is the most common treatment for localized BCC. Traditional approaches like curettage or electrodessication are supported by older studies including prospective trials with limited follow-up time. Other techniques are discussed below. According to one evidence-based review, the best results have been obtained with surgery [175].

### 8.1. Surgery

The National Comprehensive Cancer Network (NCCN) recommends clinical margins of at least 4-mm for low-risk BCC treated with standard excision with postoperative margin assessment (SEPMA) [176]. The results of Wolf et al. showed that, for well-circumscribed BCC < 2-mm in diameter, excision with 4-mm clinical margins guarantees a complete removal in more than 95% of cases [177]. In a retrospective study on 3957 consecutive excisions of BCC, primary tumors of any size on the neck, trunk, and extremities had a 5-year cure rate > 95% [178]. If SEPMA is utilized for high-risk BCC, wider surgical margins (more than 6 mm) than those reported for low-risk BCC are necessary, and greater recurrence rates of this tumor should be expected [176].

### 8.2. Mohs Micrographic Surgery (MMS)

MMS, or Mohs surgery, is a surgical procedure in which the complete excision of the NMSC is examined by microscopic margin control. MMS is the treatment of choice for high-risk and recurrent BCCs showing superior long-term cure rates than other surgical treatments. MMS allows intraoperative analysis of 100% of excision margin. Rowe et al. reported that the 5-year recurrence rates for primary and recurrent BCCs treated with MMS are 1% and 5.6%, respectively, compared with 10.1% and 17.4%, respectively, for SEPMA [179]. Likewise, the 10-year recurrence rates for primary facial BCCs were 4.4% for MMS and 12.2% for SEPMA [180]. The excision with complete circumferential peripheral and deep margin assessment (CCPDMA), using intraoperative frozen section assessment, is a valid alternative to MMS because it includes an entire assessment of all deep and peripheral margins [176]. 

### 8.3. Curettage and Electrodessication

Curettage and electrodessication is a fast and cost-effective technique for superficial lesions and recommended by the NCCN for properly selected and low-risk BCCs. However, these techniques do not allow for histologic margin assessment [176]. It has to be considered that these techniques should be avoided in areas with terminal hair growth such as the scalp, pubis, axillae, or the beard area in males, due to the risk of follicular tumor extension. Silverman et al. reported in a study of more than 2300 BCCs a 5-year recurrence rate of 3.3% (standard error [SE]: 1.5%) for lesions of any diameter localized in the L area (trunk and extremities, excluding hands, nail units, pretibia, ankles, and feet). Lesions in the M area (cheeks, forehead, scalp, neck, and pretibia), had a 5-year recurrence rate of 5.3% (SE: 2.7%) and 22.7% (SE: 7.2%), for BCCs with diameters < 10 mm or >10 mm, respectively. For BCCs in the H area (“mask areas” of the face, genitalia, hands, and feet), the 5-year recurrence rates were 4.5% (SE: 2.6%) and 17.6% (SE: 5.4%) for BCC < 6 mm or >6 mm, respectively [181].

### 8.4. Cryosurgery

Cryosurgery destroys tumor cells by freeze–thaw cycles. It is a fast and cost-effective technique but lacks histological assessment of tumor margins. Even if several large case series report cure rates of 94%–99%, this technique should be limited to superficial and low-risk BCCs [182]. As demonstrated by prospective randomized trials, a drawback of cryotherapy is the poorer cosmetic result compared to other treatment approaches [183].

### 8.5. Photodynamic Therapy (PDT)

PDT consists in the application of a photosensitizing agent, generally aminolevulinic acid (ALA), or methyl aminolevulinate (MAL), followed by irradiation with a light source. Cure rates range from 70% to 90% but it has to be considered that the reported studies have short follow-up periods [184]. Roozeboom et al. reported a 5-year recurrence rate of 30.7% (95% CI, 21.5–42.6%) for ALA-PDT and 2.3% (95% CI, 0.6–8.8%) for surgical excision (*p* < 0.0001) [185]. When stratifying for tumor thickness, the ALA-PDT cure rate was 95% for primary thin NBCCs (i.e., thickness ≤ 0.7-mm) [185]. Most articles on PDT for BCC showed high cure rates for the superficial and nodular subtype of this tumor [186,187]. Considering the nodular subtypes, cure rates are better for thinner forms [185]. Therefore, this technique should be considered mainly for superficial BCCs and for thinner nodular subtype, generally in patients affected by extensive or multifocal disease, or with multiple AKs.

### 8.6. Radiation (RT)

RT is a primary therapy indicated in patients where surgery is contraindicated or for unresectable tumors. The goal of RT is a complete eradication of the BCC with preservation of the healthy tissue. Two types of RT have been utilized for the treatment of BCC, i.e., teletherapy (external beam RT) and brachytherapy [188]. RT is mainly used in patients over 60 years of age but it is contraindicated in patients affected by genetic syndromes, like BCNS or Gorlin–Goltz syndrome, due to the higher risk to induce other malignancies caused by ionizing radiation [189]. 

RT has been compared with many other treatments for BCC in prospective RCTs. Hall et al. compared cryotherapy with superficial RT in 93 patients evaluated 2 years after treatment, reporting a 4% recurrence of the disease after RT, compared to 39% after cryotherapy [190].

Avril et al. compared RT to surgery in newly diagnosed facial BCCs. Most RT patients (55%) were treated with low dose rate interstitial brachytherapy (*n*: 173) while others (12%) received conventional outpatient teletherapy. Recurrence ≤ 4 years after treatment occurred in 0.7% of the surgery group and 7.5% of the RT group (8.8% after brachytherapy, 5% after teletherapy) [191].

### 8.7. Topical Therapies

Topical 5-fluorouracil (5-FU) 5% cream, and imiquimod 5% cream are approved for the treatment of superficial BCC [192,193,194,195,196]. In an RCT that used twice daily imiquimod 5% for 12 weeks, Geisse et al. reported a 100% histologic clearance after 6 weeks of treatment [193]. Other studies reported clearance rates of 77.9% and 80.4% for superficial BCC at a 5-year follow-up [197,198]. NBCCs showed similar results, with a 76% clinical clearance using once daily imiquimod application for 12 weeks [199]. Imiquimod 5% is also utilized for patients affected by BCNS [200,201].

An RCT showed a statistically equivalent efficacy between 5-FU and imiquimod 5% in treating superficial BCC at a 12-month follow-up [202]. Other studies with longer follow-up showed a superiority of imiquimod, with a 79.7% clearance rate at 3 years compared with 68.2% for 5-FU [203]. 5-FU is not recommended for NBCC, and evidence of its efficacy in this subtype is limited to case reports [204,205].

Topical treatments may be responsible for adverse side effects, such as erythema, swelling, and erosions, thus limiting compliance and hampering effectiveness. The use of these treatments should be limited to superficial BCCs or small BCCs localized in low-risk areas that could not be treated with other regimens [206].

### 8.8. Intralesional Therapy

Several intralesional chemotherapies have been tested for BCC treatment, such as 5-FU, interferons, interleukin-2, and bleomycin, with uneven results. Adverse events are unusual, generally dose dependent, and include local effects at the treatment site and flu-like symptoms [188,207].

### 8.9. Laser Therapy

Laser therapy has been studied for BCC treatment as both monotherapy and adjunct therapy [208]. Campolmi et al. reported a 100% histologic clearance and no recurrences over a 3-year follow-up period for superficial and NBCC treated with superpulsed carbon dioxide laser therapy [209,210]. In a retrospective study of 2719 facial BCCs treated with pulsed neodymium-based laser therapy, Moskalik et al. reported a recurrence rate of 1.8% for follow-up times ranging from 3 months to 5 years [211]. Adverse effects reported with laser therapy were reactive hyperemia, edema, scarring, and soreness [208].

## 9. Targeted Therapy: Hedgehog Pathway Inhibitors (HPI)

Vismodegib and Sonidegib are targeted oral treatments approved by the Food and Drug Administration (FDA) and by the European Medicines Agency (EMA) for the treatment of BCC, when surgery or radiotherapy are not appropriate. Specifically, Vismodegib is indicated both in locally advanced and metastatic disease, while sonidegib only in the first setting [212,213,214,215]. The mechanism of action of vismodegib and sonidegib consists in the inhibition of the HH pathway. Both HH pathway inhibitors (HPI) act as SMO inhibitors, preventing the cascade of the signal and maintaining the suppression of the transcription factors GLI. Thus, the two molecules have both a cytostatic and cytotoxic action on tumoral cells [216]. However, mechanisms of resistance to HPI have been identified, including mutations of SMO and activation of noncanonical HH pathways [217].

### 9.1. Efficacy and Safety of Vismodegib in Advanced (a) BCC

The pivotal ERIVANCE trial, a phase II multicenter, nonrandomized, two-cohort trial, has enrolled 104 patients with aBCC to receive vismodegib 150 mg once daily (Table 3). The primary endpoint of the study was ORR (Objective Response Rate): the ORR, assessed through RECIST Criteria (Central Review), was 47.6% for local advanced (la)BCC and 33.3% for metastatic (m)BCC. When assessed through RECIST criteria and Investigator Review, the ORR was 60.3% for laBCC and 48.5% mBCC. Secondary endpoints, such as the median duration of the response (mDOR) and the Progression Free Survival (PFS), assessed through Central Review, reached 9.5 months [218]. The long-term update of the study demonstrated the durability of the response, the efficacy in both patient subgroups and long-term safety. Adverse events remained consistent and discontinuation rate due to adverse effects was 21.2%. Thirty-three deaths (31.7%) were reported. However, none were related to vismodegib [219].

### 9.2. Efficacy and Safety of Sonidegib in aBCC

In 2015 the multicenter, randomized, double-blind, phase II trial BOLT evaluated efficacy and safety of sonidegib and led to the approval of the drug as a first-line treatment for laBCC [218,219] (Table 3). A total of 230 patients with aBCC were randomized 1:2 into two treatment arms. In the first arm, sonidegib was administered at the dose of 200 mg once daily, while in the second arm patients received 800 mg of the drug once daily [222]. At 30 months, sonidegib 200 mg demonstrated a better safety-risk profile. Patients receiving 200 mg of therapy had an ORR assessed through the stringent mRECIST criteria (Central Review) of 56,1% for laBCC and of 71.2% assessed through mRECIST (Investigator Review), with a mDOR and a PFS of 26.1 and 22.1 months, respectively. Seventy patients (30.4%) discontinued the therapy during the trial for AEs, such as muscle spasms, alopecia, dysgeusia, weight loss, and asthenia [220,222]. AEs were effectively managed with dose adjustments or interruptions, since sonidegib offers in label the option for dose reduction of the drug.

### 9.3. Comparison between Sonidegib and Vismodegib in aBCC

A comparison between vismodegib and sonidegib in a randomized controlled clinical trial is not available. In Europe, vismodegib is approved for the treatment of laBCC and mBCC, while sonidegib is approved for the treatment of laBCC only.

Recently the ERIVANCE and BOLT trials were considered appropriate for indirect comparison between sonidegib and vismodegib [223]. Both trials had similar baseline patient characteristics, and both used ORR by central review as the primary endpoint (Table 3).

Different criteria were adopted to assess BCC severity in the two studies. In the ERIVANCE trial, the Response Evaluation Criteria in Solid Tumors (RECIST), Version 1.0 were applied [224] while in the BOLT trial, a modified version of the RECIST, mRECIST, was adopted because it is considered to have more stringent evaluation criteria and it is more likely to detect minimal signs of disease progression.

A preplanned analysis assessed the outcomes from BOLT trial with RECIST-like criteria. The most correct match is between ORR of sonidegib by RECIST-like criteria and ORR of vismodegib by RECIST criteria at the closest follow-up time points across the studies with central review [223]. At 21-month follow-up, vismodegib RECIST ORR was 47.6%, with 22.2% complete response (CR) and 25.4% partial response (PR). At 18-month follow-up, RECIST-like ORR of sonidegib was 60.6% with 21.2% CR and 39.4% PR. Assessing efficacy data using RECIST-like criteria slightly increased sonidegib ORR (from 56.1% to 60.6%) while the number of CR increased significantly at the expense of PR. The rate of progressive disease (PD) is higher for vismodegib than for sonidegib (12.7% and 1.5%, respectively) (Table 3) [223].

In both studies, vismodegib and sonidegib showed high patient discontinuation rates: around 50% vismodegib (21.2% due to AEs, 26.0% due to patient decision, 9.8% due to physician decision); similar rates were seen for sonidegib (30.0% due to AEs, 10.0% due to patient decision, and 13.0% for physician decision) [219,220]. The most common AEs registered during treatment were dysgeusia with vismodegib and fatigue with sonidegib [218,219,220,222]. Data from both studies showed that sonidegib had an approximately 10% lower incidence of most AEs compared with vismodegib; the time to onset of AEs also indicated that patients treated with sonidegib may experience AEs slightly later than with vismodegib [223]. For the management of AEs, sonidegib is the only HPI that offers in label the option for dose modification (alternative dosing): 200 mg every other day.

## 10. BCC Prevention

Although early diagnosis and prompt treatment are indispensable means to improve BCC outcomes, the implementation of prevention measures may play a crucial role, especially if these are applied in childhood and adolescence. 

Prevention consists of lifestyle changes such as avoiding sunburns, tanning beds, and prolonged direct sun exposure between 10 a.m. and 4 p.m., as well as shade seeking, sunscreens application on the skin, physical barrier methods such as protective clothing, hats, and sunglasses. Preventive action should also be recommended for widespread professional UV exposure among outdoor workers [225]. Regular sunscreen use in childhood and adolescence seems more beneficial than in adulthood [9]. All these are practical indications that are not yet supported by high quality studies [226].

Continued long-term surveillance of these patients is also essential. The NCCN Guidelines recommend a whole-body skin examination every 6–12 months for the first 2 years after BCC diagnosis, and then at least annually for life [227]. However, patients are encouraged to practice active self-monitoring.

To date, BCC prevention includes the oral intake of a water-soluble vitamin B3 derivative, Nicotinamide (NAM), which is a component of foods like meat, fish, legumes, mushrooms, nuts, and grains [228]. It derives from tryptophan metabolism too, which accounts for 50% of its synthesis [229]. It is metabolized by the liver and secreted by the kidneys [230]. Nicotinamide plays a key role in the glycolysis pathway, producing NAD+ for ATP production to maintain cellular energy and sustain metabolic steps [231]. NAM deficiency, or pellagra, which targets organs with high cellular energy requirements, is characterized by photosensitive dermatitis, diarrhea, and dementia [229,231]. NAM-mediated photoprotection and skin cancer chemoprevention were studied at first in mice by Gensler et al. in 1997 and 1999 [232,233]. They found that topical and oral NAM prevented UV-induced immune suppression and tumor formation [232,233]. Subsequently, Damian et al. demonstrated similar effects in reducing UV immune suppression [234] and the development of NMSCs in Australian patients [235]. NAM is involved in preventing keratinocyte damage, and consequently skin cancer, by influencing several processes such as reduction of DNA damage and optimization of DNA damage response. NAM enhances the ATP-mediated repair of UV-induced DNA damage, and thus reduces both mutation rate and UV-induced immune suppression [4]. Moreover, NAM reduces UV-induced inflammation downregulating IL-6, IL-10, MCP-1, and TNF-α. NAM is considered a cutaneous immunity normalizer as counteracts UV-induced immune suppression [6]. All these beneficial actions may reduce aging-related skin changes [4] and NMSC incidence [235,236,237,238]. However, continuous NAM administration is needed to maintain its photoprotective effects [6].

In two double-blind, randomized, placebo-controlled phase two trials in Australians with sun-damaged skin and an average of more than 30 AKs at baseline, oral NAM was administered at doses of 500 mg twice daily and 500 mg once daily. Relative reductions in AK of 35% and 29% with twice daily and once daily NAM dosing have been reported, respectively, within 4 months [236]. A phase III randomized controlled trial (ONTRAC) was led on 386 Australians with a history of at least two NMSCs during the previous five years. The study showed that oral NAM (500 mg twice daily for 12 months) is safe and effective in reducing the rates of new NMSCs and AKs. The rate of new NMSCs was lower in the NAM group than in the placebo group (relative rate reduction, 23%; *p* = 0.02). Similar ranges of reduction were found for both BCC (relative rate reduction 20%, *p* = 0.120) and SCC (relative rate reduction 30%, *p* = 0.050) [235]. In addition, two small phase II randomized controlled trials showed that NAM may also obtain a chemopreventive and curative action in immune-suppressed transplant recipients. Thirty renal transplant recipients received placebo or nicotinamide 250 mg thrice daily for six months and reported reductions in AKs size and count without detectable effects on the blood levels of the immunosuppressive drugs regularly received by the patients [237]. Finally, 22 renal transplant recipients were treated with placebo or NAM 500 mg twice daily for 6 months and nonsignificant trends to reduction in new skin cancers and AKs were found, without significant increase in AEs nor significant change in blood parameters or blood pressure [238]. Therefore NAM (500 mg twice daily) should be considered a valid option for the BCC prevention, especially the secondary prevention in high risk patients with pre-existing BCC.

Systemic retinoids slow down the cell cycle and promote antitumor effects through a more efficient cellular repair of UV-induced damage [239]. They have been used as chemopreventive agents in the genetic syndromes and immunosuppressed patients. However, they displayed chemopreventive action for SCC and not for BCC [188].

Other studies suggest that pharmacological inhibition of COX-2 may hamper epithelial neoplasms, and that daily use of celecoxib might reduce the risk of developing BCC [240,241]. High-risk patients with a positive history of past BCC seem to take advantage from celecoxib treatment. However, there is poor evidence in the literature and the results are too conflicting to recommend it for chemoprevention [239].

PDT reduces the number of new cases of AK whereas it has not a clear chemopreventive effect on NMSC [242]. Only a case report on a patient with BCNS indicates that MAL-PDT to be an effective chemopreventive agent against new BCC development. However, these results need to be validated in larger studies [243].

The dietary supplements of β-carotene and selenium have also been studied, but they did not display a chemopreventive action against BCC or SCC in patients with a history of BCC [244].

## Figures and Tables

**Figure 1 biomedicines-08-00449-f001:**
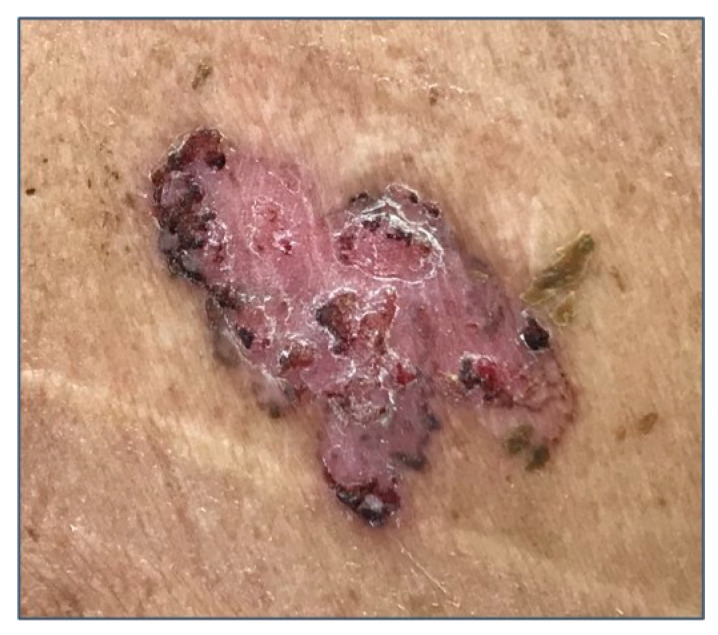
Pigmented basal cell carcinoma (BCC). Pigmented, scaly plaque with sharp, irregular edges.

**Figure 2 biomedicines-08-00449-f002:**
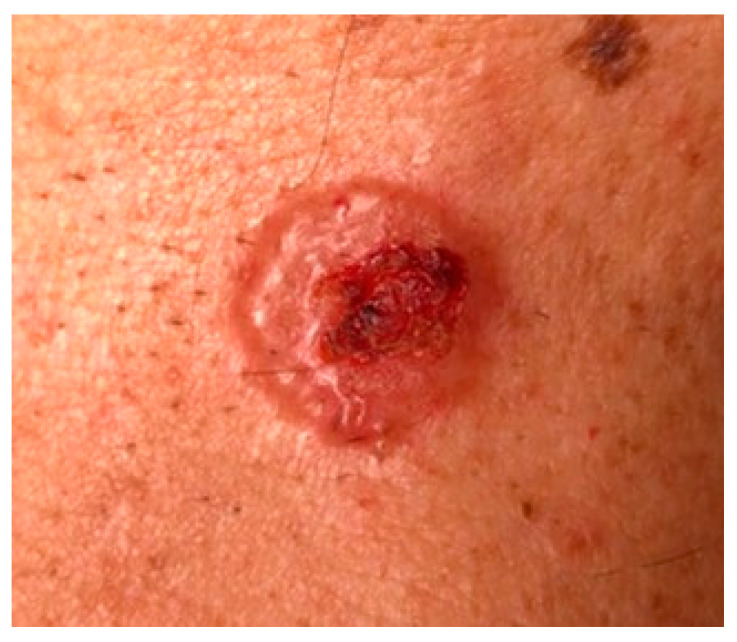
Nodular BCC. Pink, sharply delimited nodule with a characteristic shiny edge and small arborizing telangiectasias.

**Figure 3 biomedicines-08-00449-f003:**
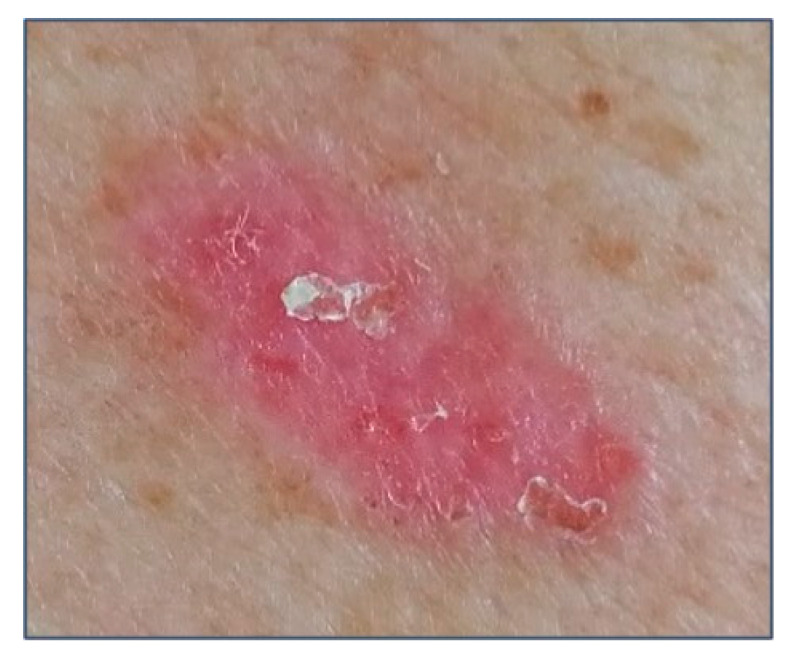
Superficial BCC. Erythematous, scaly plaque with sharp, pearly edges.

**Figure 4 biomedicines-08-00449-f004:**
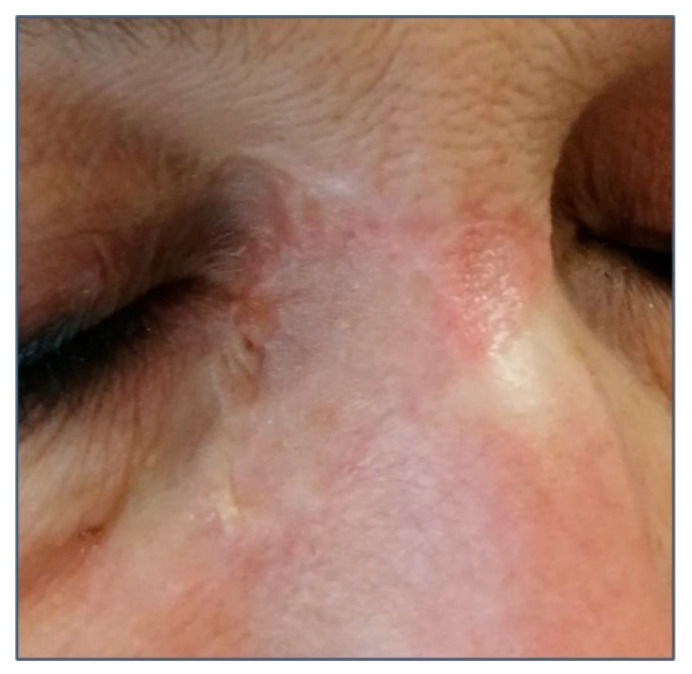
Morpheaform BCC. Poorly delimited, ivory plaque.

**Figure 5 biomedicines-08-00449-f005:**
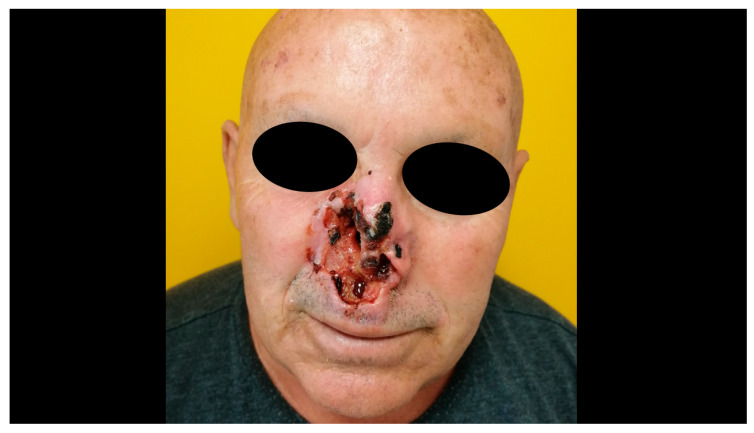
Ulcus rodens of the nasal pyramid. Extremely destructive form of BCC that shows deep tissue invasion and a high rate of postsurgical recurrence. In this case, almost the whole nasal pyramid was involved.

**Figure 6 biomedicines-08-00449-f006:**
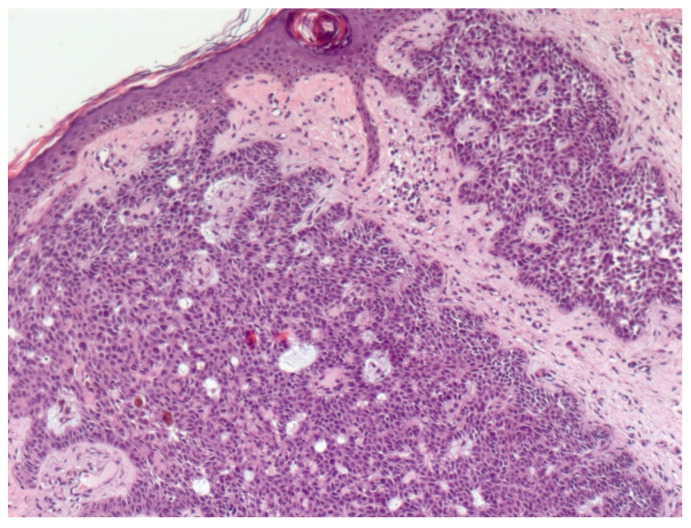
Pathological features of nodular BCC (H&E; 10×). Large basaloid lobules of different shape and size, forming a sharply delimited tumor. Typically, a peripheral palisade is shown.

**Figure 7 biomedicines-08-00449-f007:**
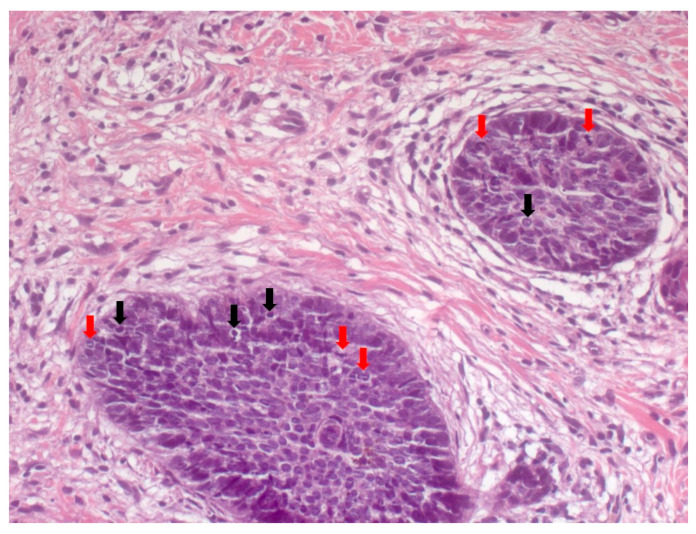
Pathological features of nodular BCC (H&E; 20×). Sharply delimited lobules with numerous mitoses (red arrows) and apoptotic figures (black arrows).

**Figure 8 biomedicines-08-00449-f008:**
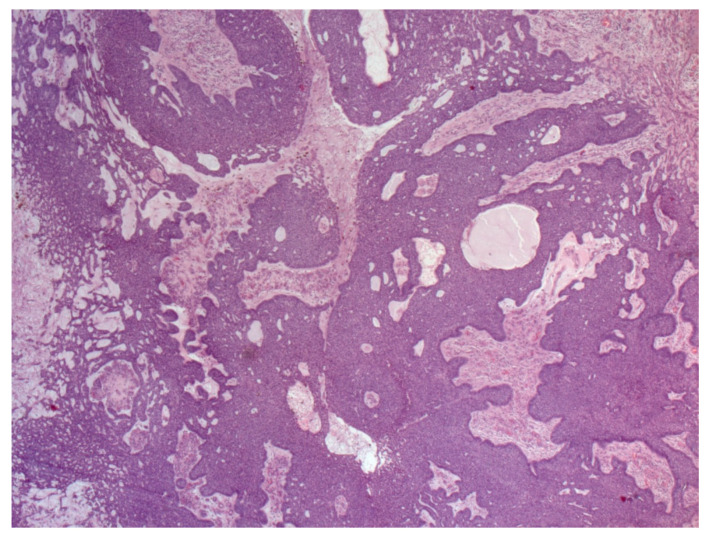
Adenoidal BCC (H&E; 2.5×). Net of basaloid cells in a mucinous stroma, mimicking a glandular formation.

**Figure 9 biomedicines-08-00449-f009:**
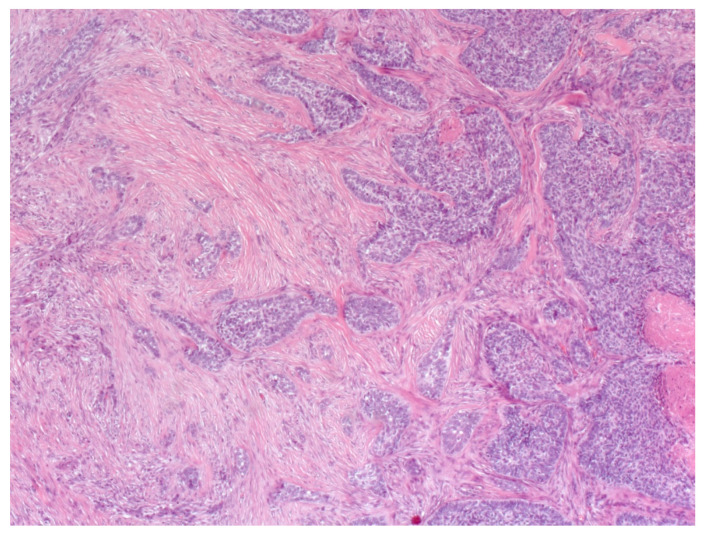
Infiltrative BCC (H&E; 5×). Small irregular clumps of basaloid cells with irregular border and limited peripheral palisading. The tumor shows extensive spread and fibrotic stroma.

**Figure 10 biomedicines-08-00449-f010:**
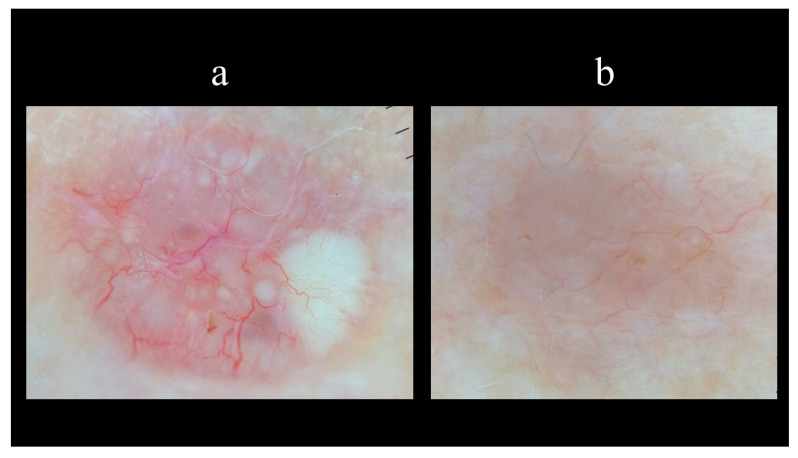
Dermoscopic features of (**a**) arborizing vessels in a 13 × 6 mm BCC, (**b**) short fine telangiectasias in a 8 × 4 mm BCC.

**Figure 11 biomedicines-08-00449-f011:**
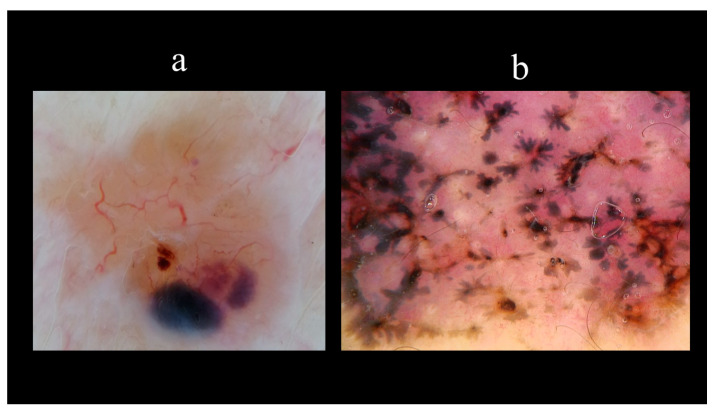
Dermoscopic features of (**a**) blue–grey ovoid nests in a 7 × 4 mm BCC, (**b**) maple leaf-like areas in a 16 × 8 mm BCC.

**Figure 12 biomedicines-08-00449-f012:**
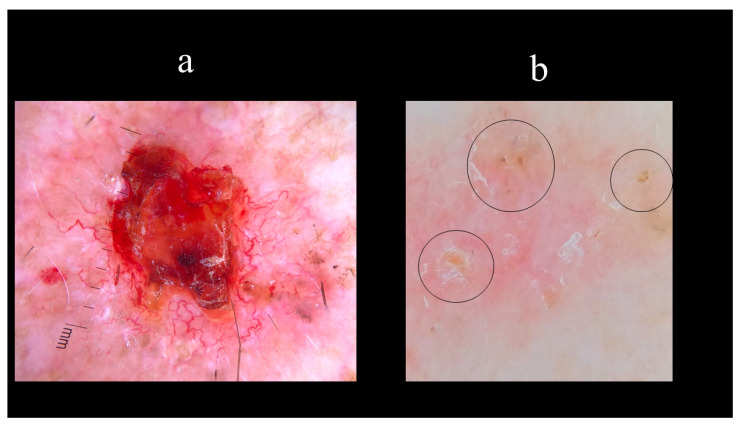
Dermoscopic features of (**a**) ulceration in a 11 × 7 mm BCC, (**b**) multiple small erosions in a 8 × 5 mm BCC.

**Figure 13 biomedicines-08-00449-f013:**
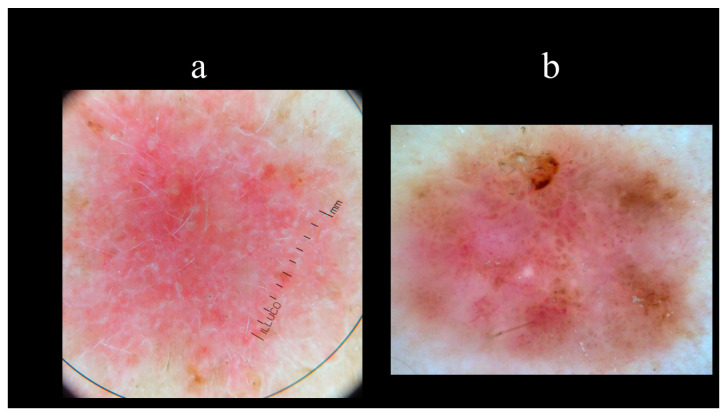
Dermoscopic features of (**a**) shiny white–red structureless areas in a 14 × 13 mm BCC, (**b**) short white streaks (chrysalis or crystalline structures) in a 14 × 7 mm BCC.

**Figure 14 biomedicines-08-00449-f014:**
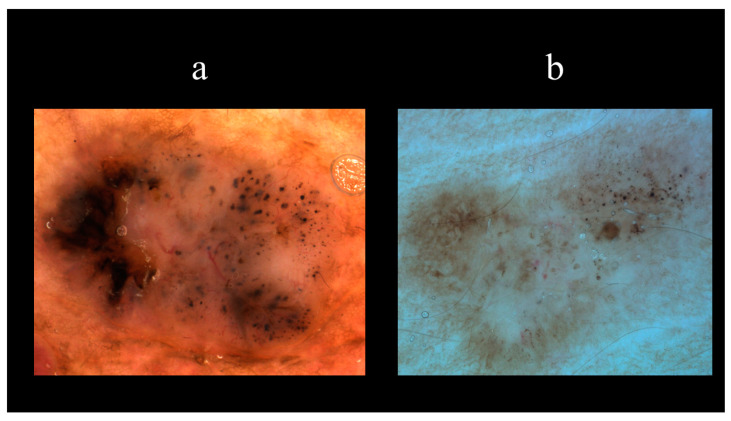
Dermoscopic features of (**a**) multiple blue–grey dots/globules in a 15 × 6 mm BCC, (**b**) in-focus dots in a 10 × 4 mm BCC.

**Figure 15 biomedicines-08-00449-f015:**
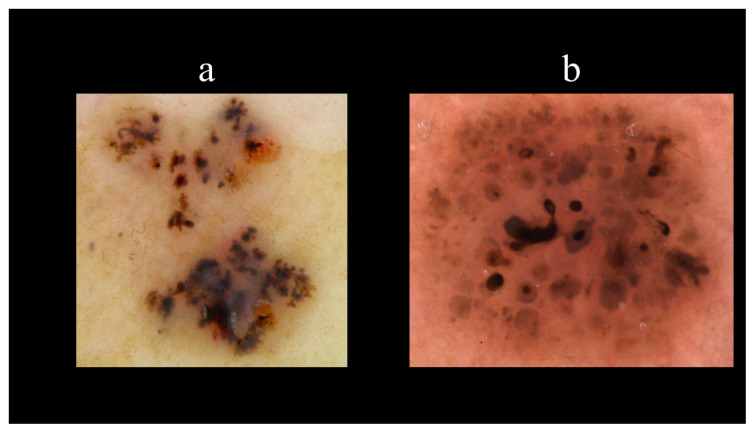
Dermoscopic features of (**a**) spoke-wheel areas in a 8 × 3 mm BCC, (**b**) concentric structures in a 10 × 10 mm BCC.

**Figure 16 biomedicines-08-00449-f016:**
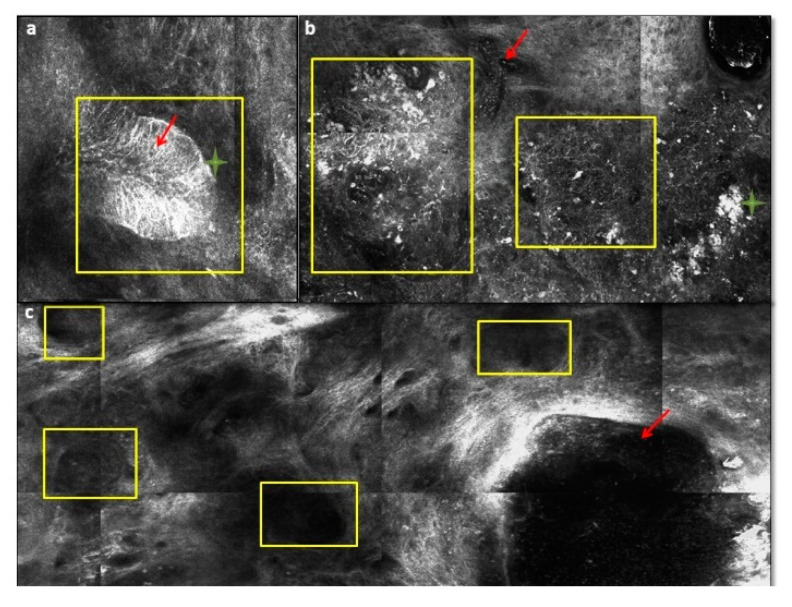
(**a**) Reflectance confocal microscopy (RCM) image of a pigmented BCC (approximately 1.5 × 1.5 mm). At the level of superficial dermis we can see a bright tumor island (yellow rectangles), which resembles an elongated cord-like structures and that is well demarcated from surrounding dermis by cleft-like dark space (green star). Fine dendritic processes are visible within the tumor island with the typical peripheral palisading of nuclei (red arrow). (**b**) RCM image of a BCC (approximately 1.5 × 1 mm) at the superficial dermis level. Canalicular vessels are curved or linear structures very frequent in BCCs with small bright cells inside which correspond to leukocytes (red arrow). Tumor islands (in the yellow rectangles) with many dendritic and plump-bright cells. Dendritic cells are melanocytes while plump-bright cells that are oval to stellate bright cells without apparent nucleus, correlate with melanophages (green star). (**c**) RCM image of BCC (approximately 1.5 × 1 mm) at the superficial dermis level. It is possible to observe many dark silhouettes (yellow rectangles), which are tumor islands in nonpigmented BCCs, surrounded by thickened collagen. On the right hand side there is a dark oval area which corresponds to an ulceration (red arrow).

**Figure 17 biomedicines-08-00449-f017:**
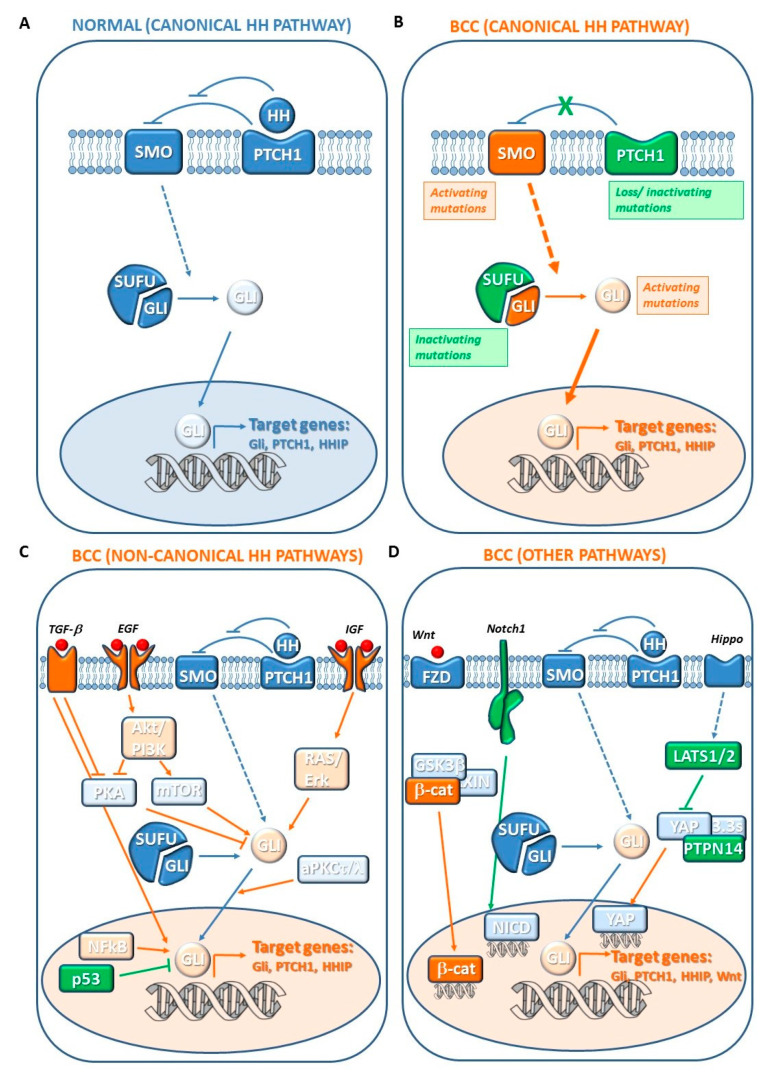
Pathways involved in BCC pathogenesis. (**A**) Canonical Hedgehog (HH) pathway is involved in epidermal physiology. Extracellular HH ligands bind to PTCH1 receptor relieving the inhibition of smoothened (SMO) by PTCH1 itself. SMO activates a signaling cascade of interacting proteins, including suppressor of fused (SUFU), resulting in activation of GLI family of transcription factors. The HH target genes include GLI1, PTCH1, and HH interacting protein (HHIP1) that regulate the pathway itself (**B**,**C**) Aberrant activation of HH pathway is a tumor-driver in BCC pathogenesis. Human BCCs may display loss or inactivating mutation in PTCH1 or SUFU, as well as activating mutations in SMO or GLI. However, activation of noncanonical HH pathways (e.g., EGFR, IGF, TGFβ pathways) may contribute to BCC development by transcription or post-translational modification of GLI bypassing the HH-mediated SMO activation. GLI activity is regulated positively by RAS, TGFβ, PI3K/AKT, NF-kB, and aPKCι/λ, and negatively by p53 and PKA. (**D**) Mutations in other genes have been implicated in BCC development, including components of WNT, Notch1, and Hippo pathways. Canonical WNT pathway is required for HH pathway-driven development of BCCs in a mouse model. WNT pathway activation has been identified in human BCCs that exhibit overexpression of WNT proteins and nuclear localization of β-catenin. Although Notch1 cannot be considered a tumor driver, its inhibition promotes tumor persistence in PTCH1 conditional mice whereas its activation induces tumor regression. In fact, human BCCs display inactivating mutation of Notch1 that is a regulator of epidermal differentiation. Inactivating mutations in LATS1/2 and PTPN14 genes of Hippo pathway have been found in human BCCs. Those mutations induce nuclear translocation of YAP and, in turn, cell proliferation. Activated or inactivated pathways are reported in orange or green, respectively [100].

**Table 1 biomedicines-08-00449-t001:** Prognostic groups of BCC according to Dandurand et al. [43].

Low Risk BCC	Intermediate Risk BCC	High Risk BCC	
Superficial primary BCC	Superficial recurrent BCC	Morpheaform or poor-defined	
Nodular primary BCC when: <1 cm in intermediate risk area <2 cm in low risk area	Nodular primary BCC when: <1 cm in high risk area >1 cm in intermediate risk area >2 cm in low risk area	Nodular primary BCC when: >1 cm in high risk area	
Pinkus tumor		Histological forms: aggressive	Recurrent forms (apart from superficial BCC)

High-risk zones are the nose, periorificial areas of the head and neck; intermediate-risk zones are the forehead, cheek, chin, scalp, and neck; low-risk zones are the trunk and limbs. Aggressive histological forms include micronodular, morpheaform, and metatypical basosquamous forms. Perineural invasion also seems to be a histological sign of aggressiveness.

**Table 2 biomedicines-08-00449-t002:** Dermoscopic structures of BCC.

Vascular Structures	Structures Related to Pigment	Nonvascular/Nonpigmented Structures
Arborizing vessels	Maple leaf-like areas	Ulcerations
Short fine telangiectasias	Spoke-wheel areas	Multiple small erosions
	Blue–grey nests and globules	Shiny white–red structureless areas
	In-focus dots	White streaks
	Concentric structures	

**Table 3 biomedicines-08-00449-t003:** Comparative efficacy of sonidegib (BOLT trial) [220] and vismodegib (Erivance trial) [218].

	Sonidegib 200 mg Daily Central Review RECIST-Like 18-Month Follow-up	Vismodegib 150 mg Daily Central Review RECIST 21-Month Follow-up
Overall response rate *n* (%); 95% CI	40 (60.6); 47.8–72.4	30 (47.6); 35.5–60.6
Complete response *n* (%)	14 (21.2%)	14 (22.2%)
Partial response *n* (%)	26 (39.4%)	16 (25.4%)
Stable disease *n* (%)	20 (30.3%)	22 (34.9%)
Progressive disease *n* (%)	1 (1.5%)	8 (12.7%)
Unknown *n* (%)	5 (7.6%)	3 (4.8%)

Another multicenter, single-arm, open-label safety trial, the STEVIE study, has enrolled 1215 patients to receive vismodegib 150 mg once daily. The primary endpoint was safety: most patients showed treatment-related side effects, including muscle spasms, alopecia, dysgeusia, weight loss, and asthenia. Secondary endpoint was efficacy: ORR assessed through RECIST 1.1 Criteria (Investigator Review) was 68.5% for laBCC and 36.9% for mBCC [221].

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
