# Peer review of "Basal Cell Carcinoma: From Pathophysiology to Novel Therapeutic Approaches"

_biomedicines, 2020, doi:10.3390/biomedicines8110449_

Round 1

Reviewer 1 Report

The manuscript by Fania et al., presents an interesting and extensive review about BCCs. Authors discuss clinical, histopathological, and dermatoscopic features in detail. In my opinion some molecular aspects might be described deeply. For example, it could be of interest to describe in the Risk factor section the role of skin and immunological aging in BCC onset and progression. In fact, several data indicate that a compromised skin integrity such us the photo-damaged skin may play an indirect role in non-melanoma skin cancer.

Hh signaling has been well reported. In contrast, Wnt signaling is very shortly described. Due to the intense and documented cross-talk between Wnt and Hyppo signaling pathway, I suggest to consider to treat WNt signaling as a merged argument with Hyppo signaling. It is not clear the reason to narrate WNt and Notch signaling in the same paragraph.  

Reviewer 2 Report

Manuscript Title – Basal Cell Carcinoma: From Pathophysiology to Novel Therapeutic Approaches

General comments

The review is well done, but several issues must be addressed before it is suitable for publication. Some minor English editing is also necessary.

Reply: We thank the reviewer for this comment. The article has been reviewed by a native English speaking that made some minor editing.

Specific comments

Abstract

1. Page 1, Line 20 – “…several effective therapeutic approaches are available for this form”. – This is not clear and should be rephrased.

2. Page 1, Line 23 – Replace “Hedgehog signaling” with “Hedgehog signaling pathway”.

Reply: We have followed these suggestions and modified these sentences.

Introduction

3. Page 1, Line 32 – Why did the authors considered actinic keratosis as skin cancer? It is considered as a premalignant lesion.

4. Page 1, Line 44 - Replace “environmental and genetic features of the patient” with “environmental factors and genetic features of the patient”.

Reply: We thank the reviewer for these comments. We have followed the suggestion number 4 and modified the sentence. Regarding actinic keratosis (AK), actually many Authors consider it as a non-melanoma skin cancer because they believe that AK should be classified as in situ squamous cell carcinomas (SCCs) (Ackerman, 2003; Heaphy and Ackerman, 2000). For this reason, we cited AK in the NMSC group.

BCC risk factors

5. Page 2, Line 8 – The authors mention “male sex” as a risk factor for BCC. This is still a controversial issue, and the authors should discuss it in more detail.

Reply: Following reviewer suggestions, this issue has been discussed.

6. Page 2, Line 11 - Why do the authors, in the previous paragraph, list the risk factors in a certain order, and now begin to describe them in detail in a different order?

Reply: Following reviewer suggestions, the list has been accordingly modified.

7. Page 2, Line 17 – “A population-based… adulthood.” Include a citation after this statement.

Reply: The citation has been added.

8. Page 2, Line 49. As previously mentioned for risk factor “male sex”, risk factor “ionizing radiation” should be discussed more thoroughly. Exposure to ionizing radiation, especially at young age, is a quite important risk factor for BCC. This was clearly shown in the tinea capitis epilation settings – Boaventura et al., Karagas et al.; Pópulo et al., Shore et al., among other reports.

Reply: Following reviewer suggestions, this issue has been discussed.

9. Page 3, Line 6 – “… after more than one NMSC…” This is not clear.

Reply: The sentence has been rewritten.

10. Page 3, Line 11 – “…increases the risk of NMSC and depends…” I believe the authors wanted to say “…increases the risk of NMSC and the increase depends…”

Reply: We apologize for the mistake. The sentence has been accordingly modified.

11. Page 3, Line 20 – “Sun exposure… sufficient” – This sentence is unclear and does not fit here.

Reply: We apologize for the mistake. The sentence has been deleted.

Clinical features and different subtypes of BCCs

12. Page 3, Line 24 - The authors do not refer mixed tumours, which are an important group, since many tumors do not have only a subtype.

Reply: As suggested, we mentioned mixed tumours.

13. Page 3, Line 34 – “…a clue to the diagnosis.” Include a citation after this statement.

Reply: We added a reference for this statement.

14. Page 4, Lines 9 to 14 – The authors do not mention that superficial BCCs are frequently multifocal. This as important aspect of superficial BCCs that may lead to its excision not always being complete. This could be discussed here or in Page 8, Line 8, when the authors mention again the superficial BCC.

Reply: As suggested, we mentioned briefly this aspect.

15. Page 6, Line 3 – The authors mention histological subtypes “morpheaform” and “infiltrating” without having previously explained what this is.

Reply: We partially do not agree with your comment. We explained the concept of morpheaform BCC on page 5 line 12. Furthermore, we removed the term “infiltrating”, because it could be considered as variant of morpheorm BCC. Cit. Scrivener Y, Grosshans E, Cribier B. variations of basal cell carcinomas according to gender, age, location and histopathological subtype. Br J Dermatol. 2002;147(1):41-7, and Wu PA. Epidemiology and clinical features of basal cell carcinoma 2014. UpToDate [internet]. Available from: http://www.uptodate.com/contents/epidemiology-and-clinical-features-of-basal-cell-carcinoma.

16. Page 6, Line 8 – “On the one hand…low risk areas.” This sentence is not clear.

Reply: We simplified the sentence removing “On the one hand…On the other hand” and adding “while” as conjunction.

Histopathological features of BCC

17. Page 8, Line 7 – “…stroma.” Make a paragraph after this sentence, since in the following sentence another subtype is mentioned.

Reply: As suggested, we made a paragraph after the sentence.

18. Differences between infiltrative and morpheaform variants should be mentioned.

Reply: As suggested, we explained the difference between the two fomes.

19. Page 8, Line 14 – “…the stroma retraction…” It is the first time the authors mention this, so they should explain what it is.

Reply: We briefly explained the concept of stroma retraction.

2. Page 9 – Explain the differences between “metatypical BCC” and “basosquamous carcinoma”, since it is not clear.

Reply: We explained briefly the differences between the two forms.

Dermoscopy

21. Page 9, Line 16 – Briefly explain what “dermoscopy” is.

22. Page 9, Line 18 – “The diagnostic… control group.” This sentence is not clear.

Reply: We have followed these suggestions and modified these sentences. Line 22 has been changed in “The diagnostic accuracy of dermoscopy for BCC could range between 95% and 99% and this could depend on BCC type compared to other lesions in the control group, such as melanocytic and nonmelanocytic lesions”.

Reflectance confocal microscopy and optical coherence tomography

23. Page 13, Line 20 – RCM limitations should be better explained.

24. Page 14, Line 21 – “…minimal use for pigmented lesions.” Please, explain this. Make a paragraph after this sentence.

25. Page 14, Line 22 – The others mention other techniques but they only state their names. Why? Are these techniques less important? More information about these techniques should be added.

Reply: We thank the reviewer for these important comments. All the sentences have been modified and better explained in the article.

Page 13, Line 20. Limitations of RCM include the imaging depth (250 µm), the limited ability to evaluate tumor invasion and deep margins, the initially steep learning curves and the cost of the instrument that represent a barrier to commercial implementation.

Page 14, Line 21. the minimal use for pigmented lesions because, in pigmented tumors, imaging techniques based on the penetration of light is more difficult.

Page 14, Line 22. Other non-invasive techniques for the diagnosis of BCC, not generally used in daily practice, include Raman spectroscopy (a spectroscopic technique typically used to determine vibrational modes of molecules), high-resolution ultrasonography, and terahertz pulse imaging (situated in the frequency regime between optical and electronic techniques)

Pathogenesis of BCC

26. Page 14, Line 32 - Replace “conserved signaling” with “conserved signaling pathway”. At the end of this sentence, include a citation.

Reply: Modified according to the reviewer suggestion.

27. Page 17, Line 7 – At the end of the figure legend include a citation.

Reply: The citation has been added.

28. Page 17, Line 19 – Replace “(PMID: 28887802)” with the appropriate reference number.

Reply: The appropriate reference has been added.

29. Page 18, Line 20 – “Mutations… BCCs” This sentence is not clear and should be rephrased.

Reply: Modified according to the reviewer suggestion.

30. Page 19, Line 28 - Replace “(PMID: 24691053)” with the appropriate reference number.

Reply: The appropriate reference has been added.

31. Page 19, Line 28 – The TERTp mutation subject should be further developed. There are more works on this subject (e.g. Pópulo et al.). TERTp mutations are important genetic alterations present in several tumours.

Reply: Following reviewer suggestions, this issue has been discussed.

32. Page 20, Line 32 – “…following high level signaling.” Does this apply to nodular and superficial subtypes? If this is the case, a coma is necessary after “epidermis”.

Reply: Modified according to the reviewer suggestion.

33. Page 20, Line 32 – “Only interfollicular… ability”. It is not clear why the progeny does not have self-renewing ability.

Reply: Modified according to the reviewer suggestion.

Treatment of localized BCC

34. Page 21, Line 3 – “…had a 5-year cure rate >95%.” With a 4-mm margin?

      Reply: The sentence has been checked and we think it is correct because the cited Authors reported: For tumors with a diameter less than 2 cm, a minimum margin of 4 mm was necessary to totally eradicate the tumor in more than 95% of cases.

35. Page 21, Line 4 – How wide should the surgical margins be?

Reply: The margins have been added in the sentence: “If SEPMA is utilized for high-risk BCC, wider surgical margins (more than 6 mm) than those reported for low-risk BCC are necessary”.

36. Page 21, Line 8 – Briefly describe MMS.

Reply: MMS has been described: “MMS, or Mohs surgery, is a surgical procedure in which the complete excision of the NMSC is examined by microscopic margin control”.

37. Page 21, Line 12 – “…(P: .10).” What does this mean?

Reply: We thank the reviewer for this comment and the (P: .10) has been removed.

Targeted therapy: Hedgehog pathway inhibitors (HPI)

38. The difference between Vismodegib and Sonidegib is not clear.

39. Page 25, Line 19 – “In fact…(P=0.020). This sentence is not clear.

Reply: Following reviewer suggestions, these sentences has been modified.

Round 2

Reviewer 2 Report

Manuscript Title – Basal Cell Carcinoma: From Pathophysiology to Novel Therapeutic Approaches

General comments

The manuscript (MS) was greatly improved; nevertheless, there are still some issues that need to be addressed.

Specific comments

 Abstract

  1. Previous comment 1 was not addressed.

Comment 1 “Page 1, Line 20 – “…several effective therapeutic approaches are available for this form”. – This is not clear and should be rephrased”

Introduction

  1. Previous comment 3 was not addressed nor did the authors reply to the question.

Comment 3. “Page 1, Line 32 – Why did the authors considered actinic keratosis as skin cancer? It is considered as a premalignant lesion.”

BCC risk factors

  1. Previous comment 5 – The authors discussed the risk factor “sex” in more detail, as suggested. I believe they could add that women can present BCCs earlier (mainly on the face and neck), because they are more concerned about their health and aesthetics, and go to the dermatologist earlier when they have lesions in these areas.
  2. Several references were added in the present version of the MS, but they were not included in the reference list. They should be included in the reference list and be cited in the text with a reference number and no as “PMID….”. The number of references in the first version and in the revised version of the MS is the same (N=221), what is not possible with the several references that had been added.
  3. Page 2, Line 16 – “Thus, aged skin characterized…” replace with “Thus, aged skin is characterized…”.
  4. Previous comment 7 – The reference added “PMID: 7857112” concerns SSC and not BCC. I believe this is not correct.
  5. Previous comment 8 – As suggested, the authors discussed the subject “ionizing radiation” in more thoroughly. I have some concerns regarding some of the added references:
  6. a) After the sentence “Several studies reported higher incidence of radiation-induced scalp BCCs in children irradiated for tinea capitis “ PMID: 1244805 – This work is very old and the authors already cited a more recent work (and more directed to skin cancer) from the same group (PMID: 6494429). I suggest removing this reference and the reference “PMID: 6494429”, keeping the reference “Shore et al. 2002 (PMID: 11893243)”. I also suggest to include “Michael D. Lichter, MD; Margaret R. Karagas, PhD; Leila A. Mott, MS; et al, 2000 Arch Dermatol. 2000;136(8):1007-1011”, and “Boaventura et al., Eur J Dermatol. 2012;22(2):225-30.”
  7. b) After the sentence “An inverse relationship 19 between BCC risk and age at radiation therapy exposure was observed” add the reference “Boaventura et al., Eur J Dermatol. 2012;22(2):225-30.”
  8. c) After the sentence “Notably, 20 the infiltrative subtype of BCC, that is considered to be more aggressive, was significantly more frequent in irradiated patients” add the reference “PMID: 24091058”.
  9. d) I believe the sentence “Specifically, mitochondria D-Loop D310 mutation rate was associated with a higher radiation dose” should be removed, since the authors do not further discuss the genetic susceptibility to irradiation.

Clinical features and different subtypes of BCCs

  1. Previous comment 15. I partially agree with the authors’ answers. The term infiltrating still appears on the Table 1 footnote. Moreover, it remains unclear the distinction between “morpheaform” and “infiltrative” subtypes (Previous comment 18).

Dermoscopy

  1. Previous comment 21. The authors added some information explaining what “dermoscopy” is, but they should also add that dermoscopy is the microscopic examination of pigmented skin lesions.

Author Response

Specific comments

Abstract

  1. Previous comment 1 was not addressed.

Comment 1 “Page 1, Line 20 – “…several effective therapeutic approaches are available for this form”. – This is not clear and should be rephrased”

Reply: Modified according to the reviewer suggestion.

Introduction

  1. Previous comment 3 was not addressed nor did the authors reply to the question.

Comment 3. “Page 1, Line 32 – Why did the authors considered actinic keratosis as skin cancer? It is considered as a premalignant lesion.”

Reply: Regarding actinic keratosis (AK), many Authors consider it as a non-melanoma skin cancer because they believe that AK should be classified as in situ squamous cell carcinomas (SCCs) (Ackerman, 2003; Heaphy and Ackerman, 2000). For this reason, we cited AK in the NMSC group. So, we believe that AK is not considered as a premalignant lesion but as an in situ squamous cell carcinoma.

BCC risk factors

  1. Previous comment 5 – The authors discussed the risk factor “sex” in more detail, as suggested. I believe they could add that women can present BCCs earlier (mainly on the face and neck), because they are more concerned about their health and aesthetics, and go to the dermatologist earlier when they have lesions in these areas.

Reply: Modified according to the reviewer suggestion.

  1. Several references were added in the present version of the MS, but they were not included in the reference list. They should be included in the reference list and be cited in the text with a reference number and no as “PMID….”. The number of references in the first version and in the revised version of the MS is the same (N=221), what is not possible with the several references that had been added.
  2. Page 2, Line 16 – “Thus, aged skin characterized…” replace with “Thus, aged skin is characterized…”.

Reply: Modified according to the reviewer suggestion.

  1. Previous comment 7 – The reference added “PMID: 7857112” concerns SSC and not BCC. I believe this is not correct.

Reply: We apology for the mistake. Actually, the PMID 7857112 corresponded to manuscript "Sunlight exposure, pigmentation factors, and risk of nonmelanocytic skin cancer. II. Squamous cell carcinoma". However, the right reference was "Sunlight exposure, pigmentary factors, and risk of nonmelanocytic skin cancer. I. Basal cell carcinoma" with PMID 7857111. Thus, we accordingly modified the number of the PMID.

  1. Previous comment 8 – As suggested, the authors discussed the subject “ionizing radiation” in more thoroughly. I have some concerns regarding some of the added references: a) After the sentence “Several studies reported higher incidence of radiation-induced scalp BCCs in children irradiated for tinea capitis “ PMID: 1244805 – This work is very old and the authors already cited a more recent work (and more directed to skin cancer) from the same group (PMID: 6494429). I suggest removing this reference and the reference “PMID: 6494429”, keeping the reference “Shore et al. 2002 (PMID: 11893243)”. I also suggest to include “Michael D. Lichter, MD; Margaret R. Karagas, PhD; Leila A. Mott, MS; et al, 2000 Arch Dermatol. 2000;136(8):1007-1011”, and “Boaventura et al., Eur J Dermatol. 2012;22(2):225-30.”

Reply: Modified according to the reviewer suggestion.

b) After the sentence “An inverse relationship between BCC risk and age at radiation therapy exposure was observed” add the reference “Boaventura et al., Eur J Dermatol. 2012;22(2):225-30.”

Reply: Modified according to the reviewer suggestion.

c) After the sentence “Notably, the infiltrative subtype of BCC, that is considered to be more aggressive, was significantly more frequent in irradiated patients” add the reference “PMID: 24091058”.

Reply: Modified according to the reviewer suggestion.

d) I believe the sentence “Specifically, mitochondria D-Loop D310 mutation rate was associated with a higher radiation dose” should be removed, since the authors do not further discuss the genetic susceptibility to irradiation.

Reply: Modified according to the reviewer suggestion.

Clinical features and different subtypes of BCCs

  1. Previous comment 15. I partially agree with the authors’ answers. The term infiltrating still appears on the Table 1 footnote. Moreover, it remains unclear the distinction between “morpheaform” and “infiltrative” subtypes (Previous comment 18).

Reply: We removed the therm "infiltrating" in the footnote of the Table I. Furthermore, we added more useful clues to differentiate morpheaform BCC from "infiltrative" BCC.

Dermoscopy

  1. Previous comment 21. The authors added some information explaining what “dermoscopy” is, but they should also add that dermoscopy is the microscopic examination of pigmented skin lesions.

Reply: Modified according to the reviewer suggestion.